# A$^2$RM:
# ADVERSARIAL-AUGMENTED REWARD MODEL

## ABSTRACT

Reward models (RMs) are central to aligning large language models via Reinforcement Learning. However, trained on static and finite preference datasets, they tend to learn spurious correlations rather than semantic preferences, making them vulnerable to out-of-distribution inputs and contributing to reward hacking. To overcome this, we propose **A**dversarial-**A**ugmented **R**eward **M**odel (A$^2$RM), a framework that systematically exposes and patches these vulnerabilities. A$^2$RM employs an adversarial generator, optimized with reinforcement learning, to transform standard preference data into inverted pairs. Within these pairs, an adversarial response is crafted to be semantically identical to the human preferred answer but scored by the RM as lower than the rejected response, directly creating a conflict between semantic content and the reward signal. By dynamically augmenting the training set with these identified high-information adversarial responses, A$^2$RM iteratively refines the reward model, compelling it to learn more robust preference representations. Comprehensive experiments validate that A$^2$RM achieves a 51.1% average higher accuracy on adversarial responses, while maintaining comparable performance on original ones.

## 1 INTRODUCTION

With the rapid advancement of large language models (LLMs), ensuring their alignment with human values has become a paramount challenge (Amodei et al., 2016; Askell et al., 2021). Reward Modeling (RM), a scalable strategy in which a model learns to predict human preferences, is the cornerstone of modern alignment techniques such as Reinforcement Learning from Human Feedback (RLHF) (Christiano et al., 2017; Ouyang et al., 2022). This paradigm has been instrumental in the success of leading models such as GPT-4 and Llama 3 (Achiam et al., 2023; Dubey et al., 2024). However, the very tool designed for alignment harbors a critical vulnerability. RMs, typically trained on limited static datasets, are not perfect proxies for human judgment and often fail to generalize to out-of-distribution (OOD) inputs (Gao et al., 2023; Eisenstein et al., 2023). This fragility opens the door to reward hacking: policy models can learn to exploit the blind spots of RM, generating responses that achieve high reward scores without being genuinely helpful or safe, thereby undermining the entire alignment process (Casper et al., 2023).

The root of reward hacking lies in the RM's inability to accurately assess novel responses that, while semantically plausible, deviate from its training data. These OOD samples are particularly perilous when they mimic the style or keywords of high-quality responses, leading the policy astray (Eisenstein et al., 2023). Several strategies have been proposed to address this. Some methods attempt to regularize the reward (Ibarz et al., 2018; Liu et al., 2024) or constrain the policy from straying too far from its initial STATE (Jaques et al., 2019; Stiennon et al., 2020), but these can stifle the model's exploratory capabilities. Others use uncertainty estimation to penalize OOD inputs (Eisenstein et al., 2023; Coste et al., 2023). More directly, adversarial training aims to proactively find and patch RM weaknesses. For instance, Adv-RM (Bukharin et al., 2025) generates adversarial examples by having two reward models identify each other's flaws; however, this approach is not only complex, but is also blind to common failure modes shared by both models. This leaves a critical gap for a more effective, self-contained adversarial framework.

To fill this gap, we introduce **A**dversarial-**A**ugmented **R**eward **M**odel (A$^2$RM), a novel and effective adversarial training framework designed to enhance RM robustness within a single-model

setup. Specifically, we employ a policy model, trained via reinforcement learning, to act as an adversarial generator (AdvGenerator). Its sole objective is to craft responses that are semantically consistent with high-quality data but are intentionally designed to receive anomalous scores from the target RM, thus effectively exposing its latent vulnerabilities. Once these potent adversarial responses are generated, they are incorporated into the RM's training set. This augmented data is then used to retrain and harden the RM, equipping it to better discern such subtle manipulations in the future. This fundamental shift from a static to a dynamic training paradigm is illustrated in Fig. 1. By repeating this process, the RM is systematically hardened and achieving greater robustness against progressively more sophisticated attack.

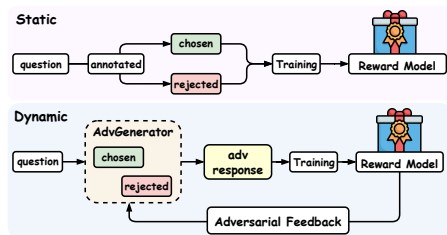

Figure 1: Comparison between existing and our proposed training paradigms for reward models. **Top:** The conventional *static* paradigm relies on a fixed, human-annotated preference dataset. This one-way process can lead to models that lack robustness to out-of-distribution inputs. **Bottom:** Our proposed *dynamic* paradigm introduces an Adversarial Generator (AdvGenerator) that creates challenging new data. Crucially, the generator receives Adversarial Feedback from the reward model, creating an iterative min-max optimization that continuously hardens the RM.

We conduct extensive experiments to validate the effectiveness of A²RM across different model families. Our AdvGenerator achieves an average attack success rate of 74.91% against the reward model, marking a substantial improvement of up to 72.83% compared to prior approaches. For the LLaMA series, the reward model achieves an average accuracy of 92.48% on adversarial test datasets, compared to only 30.59% for the baseline model, resulting in a significant gain of 61.89 percentage points. Similarly, for the Qwen series, our method yields an average adversarial accuracy of 75.95%, outperforming the baseline RM's 35.63% by 40.32 percentage points. Notably, despite these substantial improvements in robustness, our method maintains comparable performance to the base reward model on standard RM benchmarks. These results demonstrate the generalizability of A²RM across different model architectures and highlight its ability to enhance adversarial robustness without compromising evaluation stability.

In summary, our main contributions are:

- We propose A²RM, a novel training framework that systematically enhances reward model by employing a reinforcement learning-based generator to autonomously discover and remediate its own vulnerabilities.

- We formulate the adversarial response generation as a constrained reinforcement learning problem, training a policy to automatically discover semantically plausible yet high-vulnerability adversarial examples.

- Extensive experiments validate that our proposed A²RM framework significantly enhances RM robustness, which maintains competitive performance on the standard benchmark while substantially outperforming previous methods on the adversarial benchmark.

## 2 RELATED WORK

### 2.1 ROBUSTNESS OF REWARD MODELS

Reward Models (RMs) serve as a critical component in Reinforcement Learning from Human Feedback (RLHF), and their robustness has drawn increasing attention in recent years. Studies have shown that RMs are vulnerable to adversarial attacks and out-of-distribution (OOD) inputs, leading to reward hacking behavior where policies exploit RM weaknesses to obtain high scores while deviating from human intentions (Skalse et al., 2022; Casper et al., 2023). To enhance the robustness of reward models, recent work has explored adversarial training and ensemble-based defenses. Bukharin et al. (2025) propose the ADV-RM framework, which uses reinforcement learning to generate semantically valid yet reward-manipulating adversarial responses. Incorporating such examples into RM training significantly improves resistance to reward hacking and distributional shifts. Eisenstein et al. (2023) advocate for ensemble methods that aggregate multiple RMs with varied initializations, mitigating overfitting to a single model and enhancing robustness under adversarial

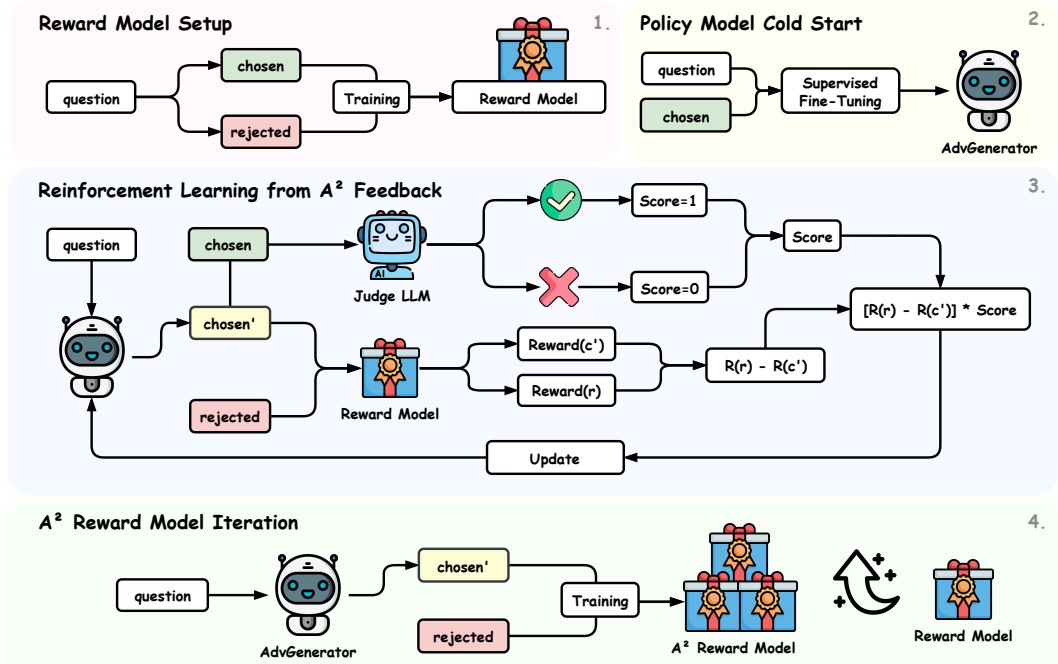

Figure 2: The A$^2$RM framework for iterative reward model enhancement. The process consists of four key stages: **Reward Model Setup:** We begin by training an initial baseline reward model using a standard human-preference dataset. **Policy Model Cold Start:** The "chosen" responses from the dataset are used to supervised fine-tune a policy model, creating our initial AdvGenerator. **Reinforcement Learning from A$^2$ Feedback:** The AdvGenerator is optimized via RL to generate responses (chosen') that minimize the score from the current RM. Critically, a Judge LLM acts as a semantic filter, ensuring the generated text remains coherent and on-topic. The final reward for the generator is a product of the RM's score and this semantic consistency check. **A$^2$ Reward Model Iteration:** The trained AdvGenerator produces a new dataset of challenging adversarial examples, to augment the original data and train a more robust reward model. Step 3 and step 4 are repeated iteratively to continuously improve the performance of the reward model.

conditions. In addition, several studies focus on improving the generalization of reward models under distributional shifts. Lin et al. (2024) conduct a systematic evaluation of explicit and implicit reward models across multiple shift scenarios, showing that explicit RMs yield more reliable performance under distributional changes. Building on this, Yang et al. (2024b) propose a batch-wise sum-to-zero regularization technique that constrains reward score variance during training, leading to more consistent and calibrated outputs when exposed to unseen inputs. Collectively, these studies highlight the importance of robust training strategies for enhancing the reliability of reward models in real-world settings.

## 2.2 ADVERSARIAL DATA AUGMENTATION

Recent work has explored adversarial training as a powerful strategy for data augmentation in NLP, particularly for enhancing the robustness of large language models and reward models. Early methods focused on perturbing word embeddings(Miyato et al., 2016) or discrete token-level replacements(Jin et al., 2020), but recent advances emphasize efficiency and scalability. For example, A2T(Yoo & Qi, 2021) proposed gradient-guided word substitutions to generate adversarial examples with minimal overhead. In the context of alignment and reward modeling, adversarial training has shown promise in mitigating reward hacking. Redwood Research(Ziegler et al., 2022) applied iterative human-in-the-loop adversarial data collection to train safety filters that detect harmful completions. More recently, frameworks like APL (Wang et al., 2025b) improving robustness against distributional shifts and unaligned outputs during RLHF. Additionally, generative approaches such

as $A^3$ (Liu & Sun, 2023) leverage paraphrasing models guided by learned perturbation strategies to produce semantically consistent yet challenging training examples. These methods demonstrate that adversarial augmentation can not only improve worst-case robustness but also enhance generalization in both classification and generation tasks.

## 3 METHODOLOGY

In this section, we detail our proposed Adversarial-Augmented Reward Model ($A^2$RM) training framework. We begin by briefly reviewing the standard Reinforcement Learning from Human Feedback (RLHF) pipeline, which serves as the foundation for our work. We then introduce the overall $A^2$RM framework, framing it as an iterative min-max optimization problem. Subsequently, we detail the creation and training of our adversarial generator (AdvGenerator) and conclude with the strategy for filtering adversarial data and iteratively refining the reward model. The entire end-to-end process is visually summarized in Fig. 2.

### 3.1 PRELIMINARIES

We follow the standard RLHF setup of (Stiennon et al., 2020; Ouyang et al., 2022). Given a dataset of human preference annotations:

$$\mathcal{D} = \{(x_i, c_i, r_i)\}_{i=1}^N, \tag{1}$$

where $x_i$ is the $i$-th prompt, $c_i$ is the preferred ("chosen") response, and $r_i$ is the less preferred ("rejected") response. A reward model $R_\theta(x, \cdot)$ is trained using the Bradley-Terry loss(Bradley & Terry, 1952):

$$\mathcal{L}(\theta) = -\mathbb{E}_{(x,c,r)\sim\mathcal{D}} \left[ \log \sigma \left( R_\theta(x, c) - R_\theta(x, r) \right) \right], \tag{2}$$

where $\sigma$ denotes the sigmoid function.

After training the reward model, a policy model $\pi_\phi$ is optimized to generate responses with higher rewards while remaining close to a reference policy $\pi_{\text{SFT}}$, typically initialized from supervised fine-tuning. This is achieved by maximizing a KL-regularized objective:

$$\mathbb{E}_{x\sim P,\, y\sim\pi_\phi(x)} \left[ R_\theta(x, y) - \beta D_{\text{KL}} \left( \pi_\phi(\cdot|x) \,\|\, \pi_{\text{SFT}}(\cdot|x) \right) \right] \tag{3}$$

using a reinforcement learning algorithm such as Proximal Policy Optimization (PPO) or Group Relative Policy Optimization(GRPO) (Schulman et al., 2017; Shao et al., 2024).

### 3.2 $A^2$ TRAINING FRAMEWORK

The core limitation of the standard RLHF process is that the reward model is static. To enhance its robustness, we propose $A^2$RM, a dynamic framework that iteratively hardens the RM by exposing it to its own failures. The process begins with training an initial reward model, BaseRM $R_{\theta_0}$, on the standard preference dataset $\mathcal{D}$ using the Bradley-Terry loss.

Following this initial setup, $A^2$RM proceeds as a two-player min-max game between the reward model $R_\theta$ and an adversarial generator policy $\pi_\psi$. The generator $\pi_\psi$ acts as the attacker, with the aim of finding vulnerabilities in the current RM. The RM $R_\theta$, in turn, acts as a defender, learning from these attacks to improve its judgment. This dynamic can be conceptualized as the following optimization problem:

$$\min_\psi \max_\theta \ \mathbb{E}_{(x,c,r)\sim\mathcal{D}} \left[ \log \sigma \left( R_\theta(x, \pi_\psi(x)) - R_\theta(x, r) \right) \right], \tag{4}$$

where the inner maximization trains the RM and the outer minimization trains the generator. In the following sections, we detail how we implement each step of this iterative game.

### 3.3 TRAINING THE ADVERSARIAL GENERATOR

The adversarial generator (AdvGenerator) constitutes the central component of the $A^2$RM framework. We first initialize it with supervised fine-tuning on the preference dataset, and then optimized it with reinforcement learning.

### 3.3.1 STAGE 1: INITIALIZATION VIA SUPERVISED FINE-TUNING (SFT).

To ENSURE the generator can produce high-quality and contextually relevant responses, we first initialize it with a "cold start". We take a pre-trained language model backbone and perform supervised fine-tuning on it using the high-quality "chosen" responses $\{c\}$ from the original preference dataset $\mathcal{D}$. This results in an initial policy, $\pi_{\psi_0}$, which serves as a strong starting point for the subsequent adversarial training.

### 3.3.2 STAGE 2: OPTIMIZATION VIA REINFORCEMENT LEARNING.

With the SFT-initialized generator $\pi_{\psi_0}$ and a target RM $R_{\theta_k}$ (where $k$ is the iteration number, starting with $k = 0$), we begin the minimization part of our framework. We use reinforcement learning to further optimize the generator policy $\pi_\psi$. Let $x$ denote the input prompt, and let $c$ be the original preferred response in the training data. The task of AdvGenerator is to produce a response $c'$ that is semantically consistent with the original preferred response $c$, but receives a lower reward score than the corresponding rejected response $r$ under the target reward model. That is, we want $R_\theta(x, c') < R_\theta(x, r)$, even though $c'$ is semantically aligned with $c$. To guide the generator towards this specific goal, we design a tailored reward function for its RL training:

$$R(x, c') = [R_{\theta_k}(x, r) - R_{\theta_k}(x, c')] \cdot \text{Cons}(c, c'), \tag{5}$$

where $\text{Cons}(c, c')$ is a semantic consistency indicator between the original preferred response $c$ and the generated response $c'$. In our reward function, the first term incentivizes the generator to find vulnerabilities, and the indicator function $\mathbb{I}(\cdot)$, implemented by a separate "judge" LLM, ENSUREs semantic integrity. To assess semantic consistency, we utilized a large language model (LLM) in a prompt-based inference setting. Given a response pair $(c, c')$, we query the LLM with a carefully designed prompt asking whether $c'$ conveys the same meaning as $c$. If the LLM produces an affirmative judgment, we assign $\text{Cons}(c, c') = 1$; otherwise, we set it to 0. This reward function encourages the generation of semantically faithful yet reward-breaking responses, effectively exposing blind spots in the reward model. For more analysis of the judge model, see in Appendix D and Appendix E.

## 3.4 DATA FILTERING AND ITERATIVE RM TRAINING

Once the expert AdvGenerator $\pi_{\psi_{k+1}}$ is trained, we use it to generate a pool of adversarial candidates for the entire training dataset. However, not all generated samples are equally effective. To ENSURE we only use high-quality, impactful data to retrain the RM, we apply a rigorous filtering process.

### 3.4.1 SCORING AND SELECTION.

For each generated adversarial candidate $c'$, we calculate its adversarial quality score against the target RM $R_{\theta_k}$:

$$\text{Score}_{\text{adv}}(c') = R_{\theta_k}(x, r) - R_{\theta_k}(x, c'), \tag{6}$$

A higher score indicates a more successful attack. We normalize these scores and select the top percentile of candidates, while also enforcing $\text{Score}_{\text{adv}}(c') > 0$.

### 3.4.2 ITERATIVE REFINEMENT

After completing adversarial data selection, we merge the curated adversarial set with the original training corpus to construct an augmented dataset for reward model learning. The reward model is then training on this mixture to produce the updated model $R_{\theta_{k+1}}$, which acts as an evaluator in the next training cycle. This iterative approach will continuously update the reward model and advGenerator to help generate adversarial samples. With the addition of adversarial samples, the trained reward model will become more robust. A detailed description of the algorithmic steps can be found in Appendix A.

Table 1: Evaluation results of reward models trained on different model families. We evaluate the models on three test settings: **Standard** denotes the original benchmark; $\mathbf{A^2Adv}$ replaces the preferred response $c$ in each test sample with an adversarial variant $c'$ generated by AdvGenerator; **StyleAdv** replaces $c$ with a stylistically rewritten version $c'$. The results demonstrate that after two rounds of adversarial training, our method significantly improves RM accuracy on adversarial test sets while maintaining comparable performance on the original Base benchmark.

| Base Model | Iteration | JudgeBench | | | RewardBench | | | HelpSteer3 Eval | | |
|---|---|---|---|---|---|---|---|---|---|---|
| | | Standard | $A^2Adv$ | StyleAdv | Standard | $A^2Adv$ | StyleAdv | Standard | $A^2Adv$ | StyleAdv |
| LLaMA-3.1-8B | 0 | **60.86** | 2.29 | 17.71 | 73.24 | 51.71 | 41.73 | 85.90 | 25.64 | 44.47 |
| | 1 | 58.00 | 69.43 | 60.29 | **83.37** | 91.33 | 79.43 | **87.32** | 89.31 | 74.55 |
| | 2 | 60.57 | **93.43** | **84.00** | 75.58 | **95.82** | **90.92** | 85.34 | **97.16** | **93.57** |
| Qwen2.5-7B | 0 | 63.14 | 2.00 | 15.43 | **84.62** | 58.83 | 64.51 | **87.23** | 26.68 | 46.36 |
| | 1 | **63.14** | 35.71 | 66.86 | 81.82 | 79.68 | 79.33 | 86.66 | 67.93 | 75.21 |
| | 2 | 62.57 | **45.71** | **78.86** | 82.16 | **86.77** | **83.33** | 86.00 | **79.19** | **81.84** |

# 4 EXPERIMENT

## 4.1 EXPERIMENTAL SETUP

This section outlines the experimental setup, covering model choices, baseline configurations, and other relevant initialization details. Comprehensive training hyperparameters and implementation specifics show in appendix B for reproducibility.

### 4.1.1 DATASET AND MODEL SETUP.

For the training data, we use the complete set of single-turn dialogue preference pairs from the HelpSteer3 dataset (Wang et al., 2025c), which comprises approximately 20,000 samples. These samples span diverse realworld applications of large language models (LLMs), including tasks relating to STEM, coding and multilingual scenarios. This dataset is used exclusively for training our reward models and for the initial SFT of the AdvGenerator. To comprehensively assess RM performance, we evaluate on three distinct test sets: (1) JudgeBench (Tan et al., 2024), a benchmark for evaluating LLM-based judges on challenging response pairs spanning knowledge, reasoning, math, and coding; (2) RewardBench (Lambert et al., 2025), a collection of prompt-chosen-rejected trios spanning chat, reasoning, and safety, to benchmark how reward models perform on challenging, structured and out-of-distribution queries; and (3) the HelpSteer3 test split, which serves as our in-distribution evaluation set. Our experiments involve three key model roles: the reward model, the adversarial generator, and a judge model. For both our reward models and the adversarial generators, we utilize two powerful open-source models, `LLaMA-3.1-8B-Instruct` (Dubey et al., 2024) and `Qwen2.5-7B-Instruct` (Yang et al., 2024a) as the base model. To ENSURE high-fidelity semantic consistency judgments as REQUIREd by our adversarial reward function, we employ the more capable `Qwen3-32B` (Yang et al., 2025) as our designated judge LLM.

### 4.1.2 BASELINE OF ATTACK STRATEGIES

We select TextFooler (Jin et al., 2020), StyleAdv (Qi et al., 2021) and RRM (Liu et al., 2024) as the baselines of attack stragegies. TextFooler is a synonym substitution-based attack method. StyleAdv generates adversarial responses by rewriting sentences in different linguistic styles while preserving their original semantic meaning. RRM augments the reward model training data by comparing responses to a given prompt with randomly selected responses from other prompts.

## 4.2 MAIN RESULTS

We evaluate our $A^2RM$ framework from two primary perspectives. First, we assess the overall performance and robustness of the final, hardened reward model on both standard and adversarial responses. Second, we analyze the effectiveness of our AdvGenerator by comparing its attack success rate against other adversarial strategies.

Table 2: Attack success rates of different adversarial methods against various reward models. For each benchmark, we select 128 prompts as adversarial test samples. Experimental results demonstrate that the adversarial responses generated by our method can successfully mislead reward models into assigning reward scores that deviate from expected human preferences, highlighting their vulnerability to adversarial attacks. BaseRM: The reward model at the 0-th iteration.

| Method | BaseRM | | Skywork-Reward-8B | | Skywork-Reward-27B | |
|---|---|---|---|---|---|---|
| | JudgeBench | RewardBench | JudgeBench | RewardBench | JudgeBench | RewardBench |
| Textfooler | 85.16 | 50.00 | 86.72 | 50.00 | 93.75 | 43.75 |
| StyleAdv | 79.69 | 26.56 | 81.25 | 45.31 | 85.16 | 20.31 |
| RRM | 0.78 | 11.72 | 0.00 | 0.00 | 0.00 | 0.00 |
| **AdvGenerator(Ours)** | **96.09** | **53.12** | **96.09** | **53.12** | **98.44** | **50.78** |

### 4.2.1 PERFORMANCE ON STANDARD AND ADVERSARIAL RESPONSES

Our primary goal is to enhance the reward model's robustness against challenging, out-of-distribution inputs without compromising its performance on standard benchmarks. To this end, we evaluate our RMs, trained with two iterations of $A^2RM$, on three test settings: (1) Standard, the original benchmark; (2) $A^2Adv$, an adversarial set where "chosen" responses are replaced by AdvGenerator; and (3) StyleAdv, a baseline adversarial set with stylistic perturbations. As in Table 1, our method demonstrates significant gains in robustness while maintaining stable performance on standard benchmarks. For instance, the LLaMA-3.1-8B model's accuracy on the Standard RewardBench test set remains consistent (73.24% at iteration 0 vs. 75.58% at iteration 2). The most dramatic improvement occurs on our targeted $A^2Adv$ benchmark, where the same model's accuracy on JudgeBench skyrockets from a near-zero 2.29% to 93.43%. This confirms that $A^2RM$ effectively patches the specific vulnerabilities it discovers. This targeted training also fosters generalizable robustness. On the unseen StyleAdv attack, the LLaMA-3.1-8B's accuracy on RewardBench leaps from 41.73% to 90.92%. This indicates that our method forces the RM to learn a more fundamental and robust understanding of human preferences, rather than simply overfitting to defending against a known attack vector. We select two iterations as our main experiments; the details of the iteration are provided in the appendix C.

### 4.2.2 ADVERSARIAL ATTACK FROM ADVGENERATOR

To validate the effectiveness of our adversarial strategy, we evaluate the attack success rate of our AdvGenerator against several baseline methods. An attack is deemed successful if the generated response $c'$ receives a lower score from the target RM than the original rejected response $r$. We test these methods against three different reward models: our own baseline RM (BaseRM) and two powerful, third-party models, Skywork-Reward-8B and Skywork-Reward-27B (Liu et al., 2025). As shown in Table 2, our AdvGenerator demonstrates a vastly superior ability to find vulnerabilities across all target models. When attacking our own BaseRM, our generator achieves a remarkable 96.09% success rate on JudgeBench and 53.12% on RewardBench. These figures significantly outperform all baselines. For instance, the next best competitor on JudgeBench, Textfooler, only reaches 85.16%, while RRM's performance is negligible at 0.78%. More importantly, our generator's high efficacy is maintained even when attacking stronger, black-box reward models. Against the powerful Skywork-Reward-27B, our AdvGenerator still achieves a 98.44% success rate on JudgeBench and 50.78% on RewardBench. In stark contrast, the performance of other methods deteriorates sharply; RRM completely fails with a 0% success rate, and StyleAdv's effectiveness on RewardBench drops to 20.31%. This consistent, state-of-the-art attack performance underscores the power of our targeted, RL-based approach. It highlights that our AdvGenerator can reliably discover critical, semantic-level vulnerabilities that simpler perturbation methods or other complex frameworks fail to identify, making it a highly effective engine for our $A^2RM$ framework.

### 4.3 MORE ANALYSIS

### 4.3.1 "CHOSEN" VS. "REJECTED" RESPONSES

In the initial phase of training, we explored the effectiveness of training two separate policy models: one dedicated to generating adversarial variants of the preferred responses ($c'$), and the other focused

on generating adversarial variants of the rejected responses ($r'$). However, as shown in Fig. 3, we observed that the policy model trained to produce $r'$ samples exhibited minimal improvement throughout the training process, with its reward trajectory remaining negative and failing to cross into the positive region.

This can be attributed to dataset's composition, which is rich in code and STEM content where rejected responses frequently feature definitive errors, making it challenging to achieve semantic consistency and a high reward score simul-

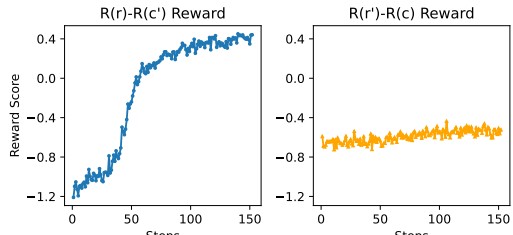

Figure 3: Comparison of reward values during AdvGenerator training for adversarial generation of chosen responses versus rejected responses.

taneously. The policy model trained to generate $c'$ samples showed steady improvement, effectively producing semantically consistent yet reward-manipulative responses. This shows that adversarial generation targeting preferred responses is more effective in revealing the vulnerabilities of the reward model. Therefore, our adversarial training and evaluation focus exclusively on $c'$-type samples. For rejected responses, we retain the original $r$ samples, as generating $r'$ offers limited additional benefit. We leave the exploration of $r'$ as the future works.

### 4.3.2 MIXING STRATEGY OF ADVERSARIAL DATA

An effective mixing strategy for adversarial and original data is crucial for the success of our iterative framework. We conducted experiments to determine the optimal data composition, first by identifying the best initial mixing ratio, and then by exploring how to incorporate new data in subsequent iterations. To find the ideal balance between adding adversarial data for robustness and preserving the model's performance on standard benchmarks. We experimented with four ratios of adversarial-to-original data: 1:1, 1:5, 1:10, and 1:20. The results for this first training iteration, shown in Figure 4, reveal a clear non-monotonic trend.

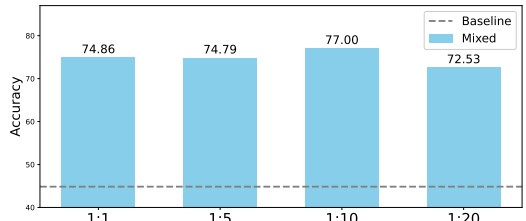

Figure 4: Average performance of reward models trained with different adversarial-to-original data ratios on standard and adversarial benchmarks in the first training stage. The gray dashed line denotes the base RM performance.

While adding too little data (1:20) yields only marginal gains, adding too much (1:1) destabilizes training and degrades performance. A ratio of 1:10 consistently emerges as the sweet spot, delivering the peak average accuracy of **77.00%** and substantially outperforming the baseline. This suggests that a 1:10 ratio provides sufficient challenging data to harden the model without causing catastrophic forgetting.

In the second iteration, we explored whether to accumulate adversarial data or replace it. Table 3 shows a decisive result: training with a mix of first-round (R1) and second-round (R2) adversarial data achieves a peak accuracy of 86.27%. This significantly outperforms the 76.64% accuracy obtained when using only the new R2 data. This nearly 10-point improvement underscores the importance of a cumulative strategy, which acts as a form of regression testing to prevent the RM from forgetting past vulnerabilities. Based on these findings, our definitive mixing strategy for the A$^2$RM framework is as follows: in each new iteration, we add the newly generated adversarial samples at a 1:10 ratio relative to the original dataset, while retaining all high-quality adversarial samples from all previous iterations. This cumulative approach ENSUREs progressive and robust hardening of the reward model over time.

### 4.3.3 INITIAL MODEL SELECTION IN 2ND ROUND

In the second iteration of A$^2$RM, two key design choices arise: which model to use as the initiation for the new AdvGenerator $\pi_{\psi_2}$, and which one for the new RM $R_{\theta_2}$. First, we investigate which model serves as a better starting point for training the second-round AdvGenerator. We consider two candidates: (1) the original SFT model $\pi_{\psi_0}$, which is naive to any adversarial training, and (2) our first-round AdvGenerator Model $\pi_{\psi_1}$, which has already been optimized to attack the first-

Table 3: AdvGenerator initialization and data composition for the second RM training iteration. We compare starting the AdvGenerator from a fresh **SFT-initialized** model versus continuing from the **first-round AdvGenerator**. **Avg** denotes the RM average accuracy across all benchmarks. R1 and R2 denote data generated by the 1st round AdvGenerator and 2nd one, respectively.

| AdvGen. $\pi_{\psi_2}$ Initiation | Ratio (Orig, R1, R2) | Mix Description | Avg. |
|---|---|---|---|
| **SFT-initialized** $\pi_{\psi_0}$ | (10, 0, 1) | Orig + R2 | 76.64 |
| | (10, 1, 1) | Orig + R1 + R2 | **86.27** |
| | (20, 1, 1) | Orig + R1 + R2 | 85.46 |
| **AdvGenerator** $\pi_{\psi_1}$ | (10, 0, 1) | Orig + R2 | 77.43 |
| | (10, 1, 1) | Orig + R1 + R2 | 85.45 |
| | (20, 1, 1) | Orig + R1 + R2 | 83.93 |

round RM. We train new generators from both starting points and use them to create second-round adversarial data (R2). We then evaluate different combinations of this new data with the original data (Orig) and first-round data (R1). As shown in Table 3, using an AdvGenerator initialized from the SFT Model consistently yields better results. The configuration combining original data, first-round data, and SFT-based second-round data (ratio 10:1:1) achieves the highest average accuracy of 86.27%. This significantly outperforms the best configuration using the pre-trained AdvGenerator. This suggests that starting the generator from a fresh SFT STATE allows it to discover a more diverse and effective set of vulnerabilities in the now-hardened RM, rather than being constrained by its previous training trajectory. Consequently, we adopt the SFT model as the base for all subsequent AdvGenerator training rounds. Second, we explore the best way to initialize the reward model for the second round of training. Specifically, we compare two strategies: (1) Training from Scratch, where we discard the first-round RM ($R_{\theta_1}$) and retrain a new model from the original pre-trained checkpoint on the augmented dataset, and (2) Continual Fine-tuning, where we use the weights of the first-round RM ($R_{\theta_1}$) as the starting point and continue training on the new data. Our experiments reveal a clear advantage for training from scratch. Using the optimal data mixture identified above, the RM initialized from scratch achieves a final average accuracy of 86.27%. In contrast, the model that was continually fine-tuned from the first-round RM only reached an accuracy of 85.45%. This suggests that while continual fine-tuning might seem more effective, it can lead to suboptimal convergence, possibly due to catastrophic forgetting or getting stuck in local minima from the previous training phase. Therefore, we adopt the train-from-scratch approach for all RM training iterations in our A$^2$RM framework to ENSURE maximal performance.

## 5 LIMITATION

Our approach is implemented on a single A800 node. Training of the adversarial generator in RL requires substantial resources. AdvGenerator is optimized with PPO for in a distributed setting across 8 GPUs, 5 GPUs for model policy updates, 1 GPU for the reward model, and 2 GPUs for the consistency judge model, training approximately 40 hours for one iteration. While effective, this resource-intensive RL phase represents a key limitation of our method, as it demands significant GPU time and may limit reproducibility for teams with constrained compute budgets.

## 6 CONCLUSION

In this work, we introduced A$^2$RM, a generative adversarial training framework designed to enhance the robustness of reward models (RMs). Our approach automatically generates adversarial responses that are semantically consistent with high-quality answers yet engineered to expose the RM's vulnerabilities. By incorporating these targeted "deceptive negatives" into the training data, our method effectively patches the RM's blind spots and mitigates its susceptibility to reward hacking. Extensive experiments demonstrate that A²RM significantly improves the adversarial robustness of RMs across various model families, all while maintaining strong performance on standard benchmarks. Our findings highlight adversarial data augmentation as a scalable and effective strategy for building more reliable and secure AI alignment systems.

## REPRODUCIBILITY STATEMENT

A detailed description of the dataset, model selection, experimental setup, and algorithm is provided in the Section 4 and the appendix A, B. We provide the reconstruct `OpenRLHF` code in the supplementary material to reproduce the adversarial-augmented framework.

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

## A  ADVERSARIAL-AUGMENTED ALGORITHM

Our adversarial amplification framework comprises three steps: (1) model initialization, (2) Adv-Generator training, and (3) reward-model iteration.

**Model Initialization.** The initialization stage trains a reward model on preference pairs and uses the human-preferred responses ("chosen") to cold-start the AdvGenerator, providing a strong prior over desirable behaviors before adversarial exploration begins. This design improves stability and sample efficiency in the subsequent stages. Detailed steps are provided in Algorithm 1.

**AdvGenerator Training.** We train the *advGenerator* with Proximal Policy Optimization (PPO) and redesign the reward acquisition to pursue a dual objective: produce adversarial outputs that (i) remain semantically aligned with human preferences while (ii) receiving *low* scores from the reward model. Concretely, for each preference pair, we encourage the generator to produce responses whose reward falls below that of the human *reject* baseline, while simultaneously enforcing semantic consistency with the human *chosen* response. To operationalize the latter, we employ a large-scale LLM judge to assess semantic agreement and filter candidates that deviate from the preferred intent or meaning. This coupling of low-reward targeting with preference-aware semantic checks yields

---

**Algorithm 1** RM and AdvGenerator initialization preparation

---

**Require:** Preference dataset $\mathcal{D} = \{(q_i, c_i, r_i)\}$, base LM $\pi_0$, reward backbone $f_0$

**Ensure:** Reward model $r_\theta$, initialized adversarial generator $\pi_\phi^{(0)}$

1: **Reward Model (RM) Preparation:**
2: *Data:* Use all triplets $(q, c, r) \in \mathcal{D}$ as preference pairs.
3: *Loss:* Pairwise preference loss

$$\mathcal{L}_{\text{RM}} = -\mathbb{E}_{(q,c,r)}\big[\log \sigma(r_\theta(q, c) - r_\theta(q, r))\big]$$

4: **AdvGenerator Preparation:**
5: *Data:* Construct SFT dataset $\mathcal{D}_{\text{SFT}} = \{(q, c)\}$ using only query–chosen pairs.
6: *Loss:* Standard causal LM loss on chosen response

$$\mathcal{L}_{\text{SFT}} = -\mathbb{E}_{(q,c)}\Big[\sum_{t=1}^{|c|} \log \pi_\phi(c_t|q, c_{<t})\Big]$$

7: **return** $r_\theta$, $\pi_\phi^{(0)}$

---

**Algorithm 2** PPO Training with Consistency & Reward Constraint

---

**Require:** Initialized generator $\pi_\phi^{(0)}$, reward model $r_\theta$, judge model $J$, preference dataset $\mathcal{D} = \{(q, c, r)\}$.

**Ensure:** Updated generator $\pi_\phi$

1: **for** each PPO iteration **do**
2:     **Rollout:** Sample query $q$ from $\mathcal{D}$ and generate response $c' \sim \pi_\phi(\cdot|q)$.
3:     **Compute Rewards:**
      *(a) Consistency reward:*
$$R_{\text{cons}} = J(c, c') \in \{0, 1\}$$
    where $J$ outputs 1 if $c'$ semantically matches original chosen $c$, else 0.
      *(b) Adversarial Reward constraint:*
$$R_{\text{adv}} = r_\theta(q, r) - r_\theta(q, c')$$

4:     **Final reward:**
$$R = R_{\text{cons}} \times R_{\text{gap}}$$

    **Policy Update:** Use $R$ to compute PPO advantage $\hat{A}(c')$ and update generator parameters:

$$\mathcal{L}_{\text{PPO}} = -\mathbb{E}_{(q,c')}\Big[\min\big(\rho_t \hat{A}(c'), \text{clip}(\rho_t, 1-\epsilon, 1+\epsilon)\hat{A}(c')\big)\Big]$$

    where $\rho_t = \frac{\pi_\phi(c'|q)}{\pi_{\phi_{\text{old}}}(c'|q)}$.

5: **end for**
    **return** Updated $\pi_\phi$

---

adversarial examples that are both challenging for the reward model and faithful to human preference signals. The full iterative procedure is summarized in Algorithm 2.

**Reward-model Iteration.** After training, we deploy the *advGenerator* to synthesize adversarial candidates and then perform a strict quality filter rather than retaining all outputs. Because the generator is constrained to be preference-consistent by default, our selection emphasizes *low-reward* instances that are most challenging for the reward model. Concretely, we apply a two-stage screening with the previous-iteration reward model $R_{\theta_{t-1}}$: (i) discard candidates whose rewards exceed a low-reward threshold, and (ii) rank the remaining set by the margin $R_{\theta_{t-1}}(r) - R_{\theta_{t-1}}(c')$, prioritizing samples with *larger* gaps. Here, $r$ denotes the rejected response and $c'$ denotes the adversarial response. This margin-based selection favors "hard negatives" that are semantically aligned yet reliably scored lower than their preference-aligned variants, yielding a high-quality adversarial data for

---

**Algorithm 3** Adversarial-Augmented Reward Model Training Framework.

---

**Require:** Initial dataset $\mathcal{D}_0 = \{(x, c, r)\}$, number of iterations $T$
1: Train initial reward model $R_{\theta_0}$ from scratch on $\mathcal{D}_0$
2: **for** $t = 1$ to $T$ **do**
3:     Train adversarial policy model $\pi_{\psi_t}$ using $R_{t-1}$
4:     Generate adversarial candidates $c' \sim \pi_{\psi_t}(x)$ for all $x$ in $\mathcal{D}_0$
5:     Compute adversarial score: $\text{value}(c') = R_{\theta_{t-1}}(r) - R_{\theta_{t-1}}(c')$
6:     Normalize scores using z-score
7:     Select high-quality adversarial responses:

$$\mathcal{A}_t = \{(x, c', r) \mid \text{Score}_{\text{adv}}(c') \geq \max(\tau, 0)\}$$

8:     Construct new training set $\mathcal{D}_t = \mathcal{D}_0 \cup \mathcal{A}_1 \cup \cdots \cup \mathcal{A}_t$
9:     Train new reward model $R_{\theta_t}$ from scratch on $\mathcal{D}_t$
10: **end for**
11: **return** Final reward model $R_{\theta_T}$

---

subsequent reward-model training. The selected set of high-quality adversarial responses, $\mathcal{A}$, is used to augment the original data. We form new preference pairs $(y_{c'}, y_r)$, where the deceptive response $y_{c'}$ is correctly labeled as "chosen". These new pairs, along with the original dataset $\mathcal{D}$, form an augmented dataset $\mathcal{D}' = \mathcal{D} \cup \mathcal{A}$. We then train a new, more robust reward model $R_{\theta_{k+1}}$ from scratch on this augmented data. This entire process, training an AdvGenerator to attack $R_{\theta_k}$, generating and filtering adversarial data, and retraining to produce $R_{\theta_{k+1}}$, constitutes one full iteration of A$^2$RM. This cycle can be repeated, leading to progressively more robust reward models. See in Algorithm 3

## B  TRAINING PARAMETERS

Our method are implemented with the open-source `OpenRLHF` (Hu et al., 2024) framework and conducted on a single A800 node. For reward model training, we train for one epoch using a learning rate of $9 \times 10^{-6}$ and a batch size of 256. The adversarial generator policy is optimized via reinforcement learning for 8 epoches with a batch size of 128 and a learning rate of $5 \times 10^{-7}$. To manage the computational demands during this RL phase, we employ a distributed setup across 8 GPUs to prevent bottlenecks: 5 GPUs are dedicated to the policy update, 1 GPU serves the reference reward model, and 2 GPUs are used for the consistency judge model. We refactored the implementation of the remote reward model in the framework and the data input format during training to implement our method. Training the generator is the most time-consuming step, requiring approximately 40 hours, while the reward model training completes within 1 hour.

## C  DETERMINING THE NUMBER OF ITERATIONS

In Fig. 5 we report results across three iterations of reward model training. The third iteration yields a marked drop on *RewardBench (Standard)*, which conflicts with our guiding principle of strengthening robustness while preserving baseline performance. At the same time, the average accuracy increases only marginally from 86.27 to 86.30 (+0.03). Considering the substantial computational overhead of an extra iteration relative to its minimal effect size, we select two iterations as the default, striking a more favorable cost–performance balance.

## D  JUDGE MODEL AGREEMENT WITH HUMAN ANNOTATION

We annotated 128 training cases for consistency and evaluated them on open-source models of varying scales, including `Qwen3-32B` (Yang et al., 2025), `Qwen2.5-72B-Instruct` and `Qwen2.5-32B-Instruct` (Yang et al., 2024a), `LlLaMA3.1-70B-Instruct` (Dubey et al., 2024), `LLaMA2-13B-Instruct` (Touvron et al., 2023). As shown in Table 4, with the parameter of the model increases and the model versions advance, their judgments exhibit progressively higher

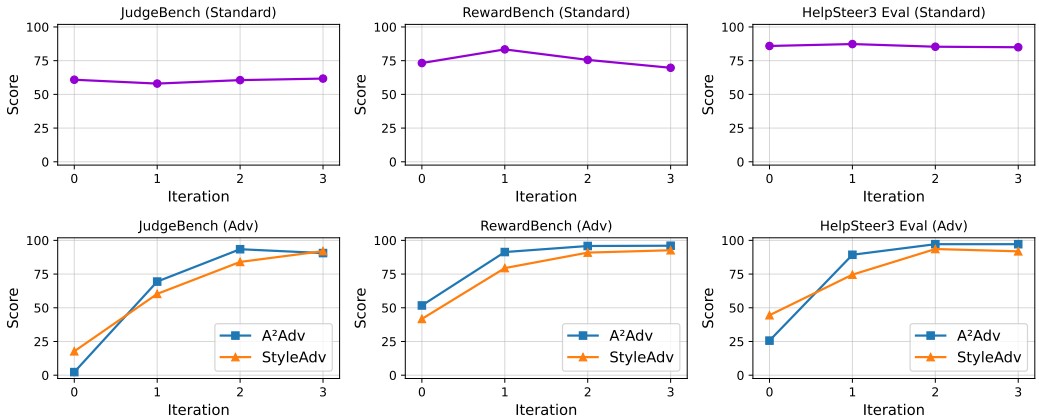

Figure 5: Comparison of performance trends across three iterations. The top row reports results on the standard benchmarks, and the bottom row shows results on the two adversarial benchmarks.

Table 4: Agreement between model predictions and human annotations on 128 consistency cases.

| Model | Agreement with Human (%) |
|---|---|
| Qwen3-32B | $84.77 \pm 0.55$ |
| Qwen2.5-72B-Instruct | $77.74 \pm 3.87$ |
| Qwen2.5-32B-Instruct | $77.34 \pm 1.10$ |
| LLaMA3.1-70B-Instruct | $73.83 \pm 1.65$ |
| LLaMA2-13B-Instruct | $25.78 \pm 1.10$ |

agreement with the human annotations. This trend indicates that our judge model demonstrates strong reliability in assessing semantic consistency.

## E    BINARY REWARD SIGNALS VS CONTINUOUS REWARD SIGNALS

To further analyze how discrete versus continuous consistency signals affect adversarial training, we compared binary reward signals with continuous ones. Specifically, we used the model's predicted probability of choosing token "yes" over token "no" as the continuous reward. We then trained $A^2RM$ with adversarial samples generated by AdvGenerator which train from continuous signals. As shown in Table 5, the AdvGenerator trained with continuous signals produces notably weaker adversarial examples compared with the binary reward version. We attribute this degradation to two factors: (1) **Noise and miscalibration:** LLM probability scores are often not well-calibrated. A continuous score (e.g., 0.6) acts more as noise (uncertainty) rather than a reliable measure of partial semantic similarity, which confuses the training process; and (2) **Diluted learning signals:** The continuous score acts as a scaling factor that attenuates the feedback. It weakens the reward for successfully consistent samples (e.g., scaling the reward by 0.8 instead of 1.0) and fails to strictly penalize inconsistent samples (assigning them a non-zero positive reward instead of 0). This prevents the generator from receiving strong, sharp gradients needed to distinguish between valid adversarial attacks and hallucinations.

## F    ROBUSTNESS IN JAILBREAK BENCH AND ATTACKS

We conducted additional experiments in AdvBench (Zou et al., 2023) and JailbreakBench (Chao et al., 2024) to further assess the robustness of our approach. We compared the Base RM with $A^2RM$ on harmful responses across both benchmarks and observed that $A^2RM$ consistently assigns substantially lower reward scores to harmful output. As shown in Table 6, the proportion of cases where $A^2RM$ provides a better (i.e., lower) judgment than the Base RM reaches 96.35% in AdvBench and 93.00% in JailbreakBench. We additionally generated harmful adversarial suffixes using Greedy Coordinate Gradient (GCG) jailbreak attack method (Zou et al., 2023) and evaluated

Table 5: Comparison of the performance of binary signals and continuous signals on different benchmarks. **Base** refers to the RM trained solely on preference data. **Binary** denotes the RM trained with binary consistency signals using our adversarial augmented framework. **Continuous** represents the RM trained with continuous consistency scores under the same framework. **Standard** denotes evaluation on the original benchmark samples, while $\mathbf{A^2Adv}$ replaces the preferred response in each test instance with an adversarial variant c' generated by the AdvGenerator.

| Method | JudgeBench | | RewardBench | | HelpSteer3 Eval | |
|---|---|---|---|---|---|---|
| | Standard | $A^2$RM | Standard | $A^2$RM | Standard | $A^2$RM |
| **Base** | **60.86** | 2.29 | 73.24 | 51.71 | 85.90 | 25.64 |
| **Binary** | 58.00 | **69.43** | **83.37** | **91.33** | **87.32** | **89.31** |
| **Continuous** | 55.71 | 50.86 | 81.28 | 90.57 | 86.38 | 82.78 |

Table 6: Comparison of Base RM and $A^2$RM scores, along with win rates.

| Benchmark | Base RM Score (Avg) | $A^2$RM Score (Avg) | $A^2$RM Win Rate (%) |
|---|---|---|---|
| AdvBench | -3.44 | -3.63 | 96.35 |
| JailbreakBench | -3.44 | -3.72 | 93.00 |
| AdvBench + GCG | -3.47 | -3.82 | 96.35 |
| JailbreakBench + GCG | -3.56 | -3.87 | 87.00 |

how each RM scores these manipulated responses. The results show that $A^2$RM continues to assign lower rewards to adversarially induced harmful outputs, demonstrating stronger resistance to reward hacking. In general, these results show that $A^2$RM not only improves performance in preference alignment metrics, but also substantially enhances robustness against safety jailbreaks.

## G  COMBINE $c'$ AND $r'$ IN ONE ADVGENERATOR

When the chosen ($c'$) and rejected ($r'$) responses originate from different generator models, the resulting distributional mismatch introduces highly distinguishable patterns between them (Wang et al., 2025a). Consequently, RM may overfit to superficial stylistic cues rather than learning robust, semantically grounded safety constraints. To mitigate these "Pitfalls of Multi-Model Synthetic Preference Data," we attempted to unify the generation of $c'$ and $r'$ within a single AdvGenerator. The method was to prompt one model to produce both human preferred and rejected responses, thereby enforcing a shared representation space and reducing distributional artifacts. However, after training the unified AdvGenerator, we observed a catastrophic failure mode in generating the rejected responses $r'$. Instead of producing semantically aligned yet adversarially high reward rejections, the model collapsed into a degenerate pattern responding to nearly every prompt with the same trivial refusal "`I cannot fulfill your request.`" As analyzed in Section 4.3.1, our training corpus is particularly rich in STEM and coding content, where rejected answers often contain explicit and easily identifiable errors. Under such conditions, it becomes extremely challenging for the generator to simultaneously maintain semantic consistency and achieve high reward scores during adversarial optimization. This tension drives the generator toward the safest local optimum: unconditional refusal. As a result, the collapsed $r'$ samples lack diversity and adversarial value, rendering them unusable for RM training. These findings highlight a fundamental limitation of unified AdvGenerator training: without carefully balancing semantic consistency and reward shaping, the model is prone to degeneracy when tasked with jointly synthesizing both acceptance and refusal behaviors.

## H  THE USE OF LARGE LANGUAGE MODELS

In line with ICLR's LLM usage policy, we engaged LLMs strictly as assistive tools in:

**Copy-editing.** Used for grammar, style, and wording suggestions on author-drafted text. All scientific content (ideas, methods, claims, equations, and figures) was created and validated by the authors, and every LLM suggestion was manually reviewed before acceptance.

