# OpenReview forum: "A$^2$RM: Adversarial-Augmented Reward Model"
_ICLR.cc/2026/Conference — Submitted to ICLR 2026_

### Official Review · Reviewer_wzXQ · 2025-10-22

**Soundness:** 3
**Presentation:** 3
**Contribution:** 2
**Rating:** 6
**Confidence:** 4

**Summary:**

This paper proposes $A^2RM$ (Adversarial-Augmented Reward Model), a framework that improves the robustness of reward models (RMs) used in RLHF. Traditional RMs trained on static preference datasets often learn superficial correlations and are vulnerable to out-of-distribution (OOD) or adversarial inputs, leading to reward hacking. $A^2RM$ introduces a reinforcement learning–based adversarial generator that creates semantically equivalent but reward-breaking responses (chosen′), verified by a judge LLM for semantic consistency. These adversarial samples are then used to augment the dataset, iteratively retraining the RM to better distinguish semantically similar but reward-deceptive cases. Experiments on LLaMA-3.1 and Qwen2.5 families demonstrate large robustness gains on adversarial benchmarks while maintaining standard performance.

**Strengths:**

1. The paper targets a concrete and consequential weakness in alignment pipelines: reward hacking, and presents a clear mechanism to directly expose and repair reward model blind spots.

2. The framework design is well-motivated and operationally practical, bridging adversarial data augmentation with preference learning.

3. Experiments are broad and results are strong, showing cross-model robustness and detailed analyses on data mixture ratios and iteration strategies that can inform future RM training protocols.

**Weaknesses:**

1. This paper aims to address the reward hacking problem but evaluates only on preference-style adversarial tests (JudgeBench, RewardBench), without connecting the results to jailbreak or safety benchmarks, where reward hacking most visibly manifests.

2. Dependence on a large external judge LLM for semantic filtering adds cost and bias risk; the paper lacks sensitivity analysis on how weaker or domain-specific judges affect results.

3. The paper assumes that semantic equivalence implies consistent reward ranking, but this can be false in practice. Two responses that are nearly semantically identical may differ substantially in reward quality, for example in code generation where a single-line bug makes one snippet incorrect while two responses are semantically identical; the paper lacks discussion addressing such counterexamples.

**Questions:**

1. How stable is the min–max optimization over multiple iterations? Does the generator collapse to trivial perturbations or continue to expose new failure modes as the RM hardens?

2. The approach heavily relies on semantic consistency judged by another LLM, how do errors or biases in that model affect adversarial quality and downstream robustness? Or did the author analyze different judge models?

3. Jailbreaking may be the most direct real-world manifestation of reward hacking. Evaluating trained models on jailbreak benchmarks (e.g., AdvBench, JailbreakBench) and some jailbreaking methods (e.g., GCG, AutoDAN or PAIR) would critically verify whether the proposed robustness extends beyond semantic perturbations to true safety-critical exploits.

---

> ### Author Response · Authors · 2025-11-20
> **Response to Reviewer wzXQ (Part 1/2)**
>
> Thank you for your insightful feedback. We appreciate your recognition of our framework’s motivation and practicality. We have conducted additional experiments on safety benchmarks and sensitivity analyses to address your concerns.
>
> ---
>
> > **Weakness 1  & Question 3:** Need connecting the results to jailbreak or safety benchmarks.
> >
>
> Thank you for this suggestion. We agree that connecting our results to safety-oriented adversarial evaluations is important. In response, we conducted additional experiments on AdvBench and JailbreakBench. We compared the Base RM with our A$^2$RM on harmful responses from both benchmarks. Across samples, the A$^2$RM assigns substantially lower reward scores to harmful outputs. The proportion of cases where A$^2$RM provides a better(lower) judgment than Base RM reaches 96.35% on AdvBench and 93.00% on JailbreakBench.
>
> |  | Base RM Score avg | A$^2$RM Score Avg | Better proportion |
> | --- | --- | --- | --- |
> | AdvBench | -3.44 | -3.63 | 96.35 |
> | JailbreakBench | -3.44 | -3.72 | 93.00 |
>
> We further generated harmful adversarial suffixes using GCG and evaluated how different RMs score these manipulated responses. A$^2$RM consistently assigns lower rewards to adversarially induced harmful outputs, demonstrating improved resistance to reward hacking attacks.
>
> |  | Base RM Score avg | A$^2$RM Score Avg | Better proportion |
> | --- | --- | --- | --- |
> | AdvBench | -3.47 | -3.82 | 96.35 |
> | JailbreakBench | -3.56 | -3.87 | 87.00 |
>
> We agree that robustness against reward hacking should extend to safety-critical scenarios. To verify this, we evaluated our models on AdvBench and JailbreakBench, as well as against GCG-based adversarial suffixes. Since these benchmarks consist of harmful prompts, a robust Reward Model should assign lower scores.
> On the standard AdvBench and JailbreakBench, our A$^2$RM consistently assigns lower rewards to harmful outputs compared to the Base RM. Specifically, A$^2$RM penalizes harmful responses more strictly than the Base RM in 96.35% of AdvBench cases. Under GCG-generated adversarial suffix attacks, A$^2$RM also keep a lower scores (–3.82 vs. –3.47).
>
> |  | Base RM Score (Avg) | A$^2$RM Score (Avg) | A$^2$RM Win Rate |
> | --- | --- | --- | --- |
> | AdvBench | -3.44 | -3.63 | 96.35 |
> | JailbreakBench | -3.44 | -3.72 | 93.00 |
> | AdvBench + GCG | -3.47 | -3.82 | 96.35 |
> | JailbreakBench + GCG | -3.56 | -3.87 | 87.00 |
>
> These results confirm that A$^2$RM does not just improve performance on preference metrics but also significantly enhances the model's resistance to safety jailbreaks.
>
> ---
>
> > **Weakness 2  & Question 2:**  Dependence on external judge LLM for semantic filtering adds cost and bias risk.
> >
>
> Thank you for raising this meaningful concern. We address the concern regarding the reliability of the Judge model from three perspectives: adherence to community standards, empirical human validation, and scalability analysis.
>
> - **Alignment with Existing Work:** We followed the methodology of recent studies [1-3] by employing an LLM-as-a-Judge for semantic evaluation. This approach has become a common choice in the research community, offering a practical balance between scalability and evaluation cost.
> - **Human Verification:** To empirically validate the reliability of this approach in our specific context, we conducted a human evaluation on 128 randomly sampled pairs of original and adversarial responses. We compared the judgments of our Judge model (Qwen3-32B) against human annotations. The results show a high agreement rate of 84.77% (± 0.55%). This high consistency, coupled with a narrow fluctuation range across repeated trials, confirms that our Judge model serves as a reliable proxy for human semantic evaluation.
> - **Evolutionary Trend & Robustness:** We further evaluated five different open-source models to analyze how judge quality varies with model capability. As shown in the table below, agreement with human annotators improves significantly with model scale and iteration.
> This trend suggests that our framework is future-proof: as open-source LLMs continue to improve, the reliability of the semantic filter and consequently the robustness of the trained A$^2$RM will naturally increase. We also believe that our framework could derive further benefits from more powerful proprietary models (e.g., GPT-5). However, given the substantial costs associated with large-scale iterative training, we prioritized open-source solutions in this work to ensure cost-effectiveness.
>
> | Model | Agreement with Human |
> | --- | --- |
> | qwen3-32B (Used) | 84.77 ± 0.55 |
> | qwen2.5-72B-Instruct | 77.74 ± 3.87 |
> | qwen2.5-32B-Instruct | 77.34 ± 1.10 |
> | llama3.1-70B-Instruct | 73.83 ± 1.65 |
> | llama2-13B-Instruct | 25.78 ± 1.10 |
>
> [1] From Generation to Judgment: Opportunities and Challenges of LLM-as-a-judge. EMNLP 2025.
>
> [2] Generative Judge for Evaluating Alignment. ICLR 2024.
>
> [3] Judging LLM-as-a-Judge with MT-Bench and Chatbot Aren. NeurIPS 2023.

---

> > ### Comment · Reviewer_wzXQ · 2025-11-26
> >
> > Thank you for the detailed response, which resolves most of my concerns. I have one additional question out of curiosity: This paper \[1\] demonstrates that when chosen and rejected responses originate from different generator models, the resulting distribution mismatch leads to highly distinguishable patterns between the two, allowing models to exploit superficial cues rather than internalizing robust safety constraints. There is also a quite similar setting about using original chosen data from dataset and paired with a self-generated response. In A²RM, the adversarially generated c′ comes from the adversarial generator, while r comes from original preference dataset. This creates a similar “Pitfalls of Multi-Model Synthetic Preference Data”. I am curious would A²RM perform better or be more robust if both c′ and r were generated by the same model. (This question will not affect my overall evaluation of the paper.)
> >
> > Reference :
> > \[1\] Wang, Yifan, Runjin Chen, Bolian Li, David Cho, Yihe Deng, Ruqi Zhang, Tianlong Chen, Zhangyang Wang, Ananth Grama, and Junyuan Hong. "More is Less: The Pitfalls of Multi-Model Synthetic Preference Data in DPO Safety Alignment." _arXiv preprint arXiv:2504.02193_ (2025).

---

> > > ### Author Response · Authors · 2025-11-28
> > > **Response to Reviewer wzXQ**
> > >
> > > Thank you for your thoughtful question. We are pleased that our previous response resolves most of your concerns, and we appreciate the opportunity to further clarify this point.
> > >
> > > > **Question:** Can both $c'$ and $r$ come from the same AdvGenerator ?
> > > >
> > >
> > >
> > > In our framework, however, the adversarial training data does not rely solely on adversarially generated $(c', r)$ pairs. The majority of samples used to train the reward model (RM) still come from the original human preference dataset, where the chosen and rejected responses share a consistent distribution. The adversarial pairs constitute only a small fraction of the total training corpus. Therefore, we believe that the training data of RM in our method does not creates a similar “Pitfalls of Multi-Model Synthetic Preference Data”.
> > >
> > > In the early stage of our experiments, we attempted to generate $r'$ using the AdvGenerator (Same training method as $c'$) and construct adversarial preference pairs $(c', r')$ for RM training. However, upon inspection of the generated $r'$ samples, we found that they failed to simultaneously achieve high reward scores and maintain semantic consistency. As discussed in Section 4.3.1, our training corpus is rich in STEM and coding content, where rejected responses typically contain explicit factual or logical errors. Under such conditions, it becomes inherently difficult for the generator to produce adversarial $r'$ responses that are both semantically coherent and adversarially valid, thus undermining the intended effect of the synthetic rejection samples.
> > >
> > > To address your suggestion of generating both $c'$ and $r$ with a single model to potentially improve robustness, we further explored a prompt driven switching method where a single AdvGenerator uses specific prompts to generate either $c'$ or $r'$ responses. Unfortunately, this setting led to a collapse mode during training: the model consistently produced the same refusal response for nearly all prompts, such as *"I cannot fulfill your request."* when asked to generate rejected outputs. Such degenerate responses offer no useful adversarial signal, and incorporating them into RM training would therefore be meaningless.
> > >
> > > We agree with your point regarding the risks of multi-model preference synthesis. Nevertheless, in A$^2$RM, adversarially generated data constitutes only a minor portion of the training set, and the dominant signal still comes from authentic human preference pairs. We have included the relevant discussion in the appendix, and we hope it further clarifies our design choices and address your curiosity.
> > >
> > > Thank you again for your constructive comments and your deeply discussion of our work.

---

> ### Author Response · Authors · 2025-11-20
> **Response to Reviewer wzXQ (Part 2/2)**
>
> > **Weakness 3 :** Lacks discussion addressing counterexamples about llm judge.
> >
>
> This is an excellent point. You are correct that in precise domains like coding, a subtle change (e.g., a one-line bug) creates a functional disparity despite visual or semantic similarity, warranting a large reward drop. Our framework handles this through two mechanisms:
>
> - **Functional Consistency Definition:** Our Judge LLM is prompted to evaluate meaning and intent, not just surface-level phrasing. For code, a capable Judge (like Qwen3) detects that a buggy function does not achieve the same intent as the correct one, thus labeling it as inconsistent (0). This prevents such "false positive" pairs from entering the training set.
> - **Binary Filter Protection:** This is exactly why we prefer a Binary (0/1) filter over a continuous one. A continuous score might assign "0.9 similarity" to a buggy code snippet, confusing the RM. Our binary filter strictly rejects it (0), ensuring that the RM is only trained on pairs that are truly functionally equivalent. Actually, using a continuous reward is less effective as the the binary one since more noise is introduced.
>
> Therefore, counterexamples like bugs are filtered out by the Judge and do not become training noise.
>
> ---
>
> > **Question 1 :** The stable of AdvGenerator in min–max optimization over multiple iterations.
> >
>
> You are absolutely correct that without intervention, the min-max optimization tends to saturate or collapse. We observed that if the generator is continually fine-tuned across iterations, it quickly overfits to the specific artifacts of the current RM, resulting in trivial or repetitive perturbations that fail to provide new learning signals.
>
> To address this stability challenge and prevent mode collapse, we implemented a "Reset" strategy: we re-initialize the AdvGenerator from its original SFT state ($\pi_{\psi_0}$) at the beginning of each new iteration.
> Our experiments (Section 4.3.3) confirm that this mechanism is crucial. By resetting the policy, we force the generator to escape local minima and explore the search space anew. This ensures that it continues to discover novel and diverse failure modes as the RM hardens, thereby maintaining a healthy and effective optimization loop.

---

> ### Author Response · Authors · 2025-11-25
> **Additional Experiment for Judge Model**
>
> > **Weaker model as judge.**
> >
>
> We employ **Llama2-13B-Instruct** as our judge model, which achieves only **25.78 ± 1.10** human alignment consistency, to examine how judge accuracy influences our adversarial augmentation framework. Experimental results show that adversarial samples generated by AdvGenerator trained with this weaker judge model lead to substantially poorer RM performance on A$^2$Adv bench compared to samples generated using a stronger judge model. This suggests that the reliability of the judge critically affects the quality of adversarial data. We believe that as model capabilities continue to advance and judge models become more accurate, A$^2$RM will exhibit even stronger robustness.
>
> |  | JudgeBench | RewardBench | HelpSteer3 Eval |
> | --- | --- | --- | --- |
> |  | standard / A$^2$Adv | standard / A﻿$^2$﻿﻿Adv | standard / A$^2$Adv |
> | base | **60.86** / 2.29 | 73.24 / 51.71 | 85.90 / 25.64 |
> | A$^2$RM with **Qwen3-32B** as judge | 58.00 / **69.43** | **83.37** / **91.33** | **87.32** / **89.31** |
> | A$^2$RM with **Llama2-13B-Instruct** as judge | 57.43 / 28.57 | 82.90 / 70.90 | 86.85 / 59.22 |

---

### Official Review · Reviewer_FLFU · 2025-10-29

**Soundness:** 3
**Presentation:** 3
**Contribution:** 3
**Rating:** 6
**Confidence:** 4

**Summary:**

The paper addresses the critical vulnerability of Reward Models (RMs) to "reward hacking," which stems from their training on static and finite preference datasets. It proposes a novel framework, $A^2RM$ (Adversarial-Augmented Reward Model), to dynamically and iteratively harden these models. The core of this method is an "AdvGenerator," a policy optimized with reinforcement learning, tasked with creating "inverted pairs". These are adversarial responses ($c'$) that are semantically identical to the human-preferred "chosen" response ($c$) but are scored by the target RM as being worse than the "rejected" response ($r$). To ensure semantic integrity, this generation process is constrained by a "Judge LLM" that filters for semantic consistency. By augmenting the training data with these challenging, high-information samples, $A^2RM$ compels the RM to learn more robust preference representations. The experimental results are strong, demonstrating that this method significantly improves accuracy on adversarial datasets (e.g., a 51.1\% average gain) while successfully maintaining performance on standard, in-distribution benchmarks. In general. I think this paper could be accepted if the authors fix some the issues.

**Strengths:**

The paper's primary strength is its new and highly effective methodology. The $A^2RM$ framework is an elegant solution to the static data problem, intelligently formulating the hardening process as a min-max optimization. The design of the adversarial reward function (Eq. 5) is particularly noteworthy, as it precisely optimizes for the desired "inverted pairs" by combining the RM's score difference with a semantic consistency check from a Judge LLM. This method is supported by comprehensive and convincing empirical evidence. First, the results in Table 1 show a dramatic improvement in robustness on both seen ($A^2Adv$) and unseen (StyleAdv) adversarial benchmarks, without the common trade-off of sacrificing performance on standard evaluations. Second, the ablation studies presented in Section 4.3 are exceptionally thorough and provide valuable, practical insights for the research community. These include key findings on the optimal 1:10 data mixing ratio, the importance of a cumulative data strategy (retaining samples from all previous iterations), and the superior performance of retraining both the RM and the AdvGenerator from their original (scratch/SFT) states in each iteration.

**Weaknesses:**

Despite its strengths, the paper has several weaknesses that should be addressed.
1) The most critical weakness is the framework's core dependency on an external, and presumably more powerful, "Judge LLM" (Qwen3-32B) to provide the semantic consistency signal $Cons(c,c^{\prime})$. This introduces a significant, undiscussed single point of failure: if the Judge LLM is itself biased, flawed, or vulnerable, it could poison the augmentation dataset and compromise the entire training process. This reliance effectively "transfers" the robustness problem from the RM to the Judge LLM. It's better to provide a experiment results on LLM judgement vs human labelling.

2) Another major weakness is the lack of analysis regarding the inconsistent attack performance shown in Table 2. The AdvGenerator demonstrates a near-perfect attack success rate on JudgeBench (e.g., 96.09\%) but a substantially lower rate on RewardBench (e.g., 53.12\%); the paper reports these figures but fails to investigate or explain this significant discrepancy.

3) Finally, the paper's claims of being an "efficient" framework are unsubstantiated. It criticizes prior work as "computationally expensive" but later admits its own RL-based generator requires 40 hours of training per iteration on a powerful 8-GPU node, yet provides no direct cost comparison to justify its efficiency claims vs other approaches. It will be better to provide more experiment results on resource and time cost.

**Questions:**

1) The Judge LLM provides a binary (0 or 1) consistency score. Would using a "soft" reward (e.g., a continuous semantic similarity score) provide a more stable and informative gradient for the AdvGenerator's RL training? If a binary reward is better, please explain why.

2) The finding in Sec 4.3.3 that retraining the AdvGenerator from its initial SFT state ($\pi_{\psi_0}$) is superior to continual fine-tuning ($\pi_{\psi_1}$) is interesting. Does this imply that the AdvGenerator "overfits" to the specific vulnerabilities of a single RM iteration, thus requiring a reset to effectively discover the \textit{new} vulnerabilities in the hardened $R_{\theta_1}$?

3) The data in Appendix C (Fig. 5) suggests that while adversarial accuracy saturates, standard benchmark performance on RewardBench begins to degrade by the third iteration. Does this point to a fundamental trade-off, where excessive adversarial training eventually harms in-distribution generalization?

4) How do the authors respond to the critique that $A^2RM$ does not solve the robustness problem but merely "transfers" it to the external Judge LLM? What would happen to the framework's effectiveness if an attacker were to target the Judge LLM itself?

---

> ### Author Response · Authors · 2025-11-20
> **Response to Reviewer FLFU (Part 1/2)**
>
> Thank you for your encouraging feedback and for recognizing the elegance and effectiveness of our A$^2$RM framework. We have carefully addressed your concerns regarding the Judge model, experimental analysis, and efficiency below.
>
> ---
>
> > **Weakness 1 :**  Results on LLM judgement vs human labelling.
> >
>
> We appreciate this suggestion.  We argue that employing an LLM-as-a-Judge is widely adopted for its balance between scalability and evaluation quality [1-3]. While we agree with your concern that a flawed judge could poison the training data, we believe that state-of-the-art LLMs have evolved sufficiently to minimize such defects. To validate this, we conducted a human-labeling study on 128 randomly sampled cases. As shown in the table below, our chosen judge (Qwen3-32B) achieved a high agreement rate of 84.77% (± 0.55%) with human annotators. This empirical evidence confirms that the judge model is reliable enough to provide high-quality supervision.
>
> | Model | Agreement with Human |
> | --- | --- |
> | qwen3-32B (Used) | 84.77 ± 0.55 |
> | qwen2.5-72B-Instruct | 77.74 ± 3.87 |
> | qwen2.5-32B-Instruct | 77.34 ± 1.10 |
> | llama3.1-70B-Instruct | 73.83 ± 1.65 |
> | llama2-13B-Instruct | 25.78 ± 1.10 |
>
> We further evaluated five different open-source models. The results reveal a clear positive correlation between model capability and judgment accuracy. This trend suggests that the reliability of our framework is future-proof and will naturally increase as the underlying base models continue to evolve. We also believe that our framework could derive further benefits from even stronger proprietary models (e.g., GPT-5). Although we prioritized open-source solutions in this work due to cost considerations, the framework is designed to scale with the best available models.
>
> [1] From Generation to Judgment: Opportunities and Challenges of LLM-as-a-judge. EMNLP 2025.
>
> [2] Generative Judge for Evaluating Alignment. ICLR 2024.
>
> [3] Judging LLM-as-a-Judge with MT-Bench and Chatbot Aren. NeurIPS 2023.
>
> ---
>
> > **Weakness 2 :** Lack of analysis regarding the inconsistent attack performance (JudgeBench vs RewardBench).
> >
>
> Thank you for this finding. We investigated the composition of the two benchmarks to understand this discrepancy:
>
> 1. **JudgeBench:** Primarily consists of **Objective Tasks** (Knowledge, Reasoning, Math, Coding). In these domains, "semantic equivalence" is binary and clear-cut (e.g., two code snippets either perform the same function or they don't). This clarity makes it easier for the generator to craft valid adversarial samples that pass the Judge.
> 2. **RewardBench:** Contains a large proportion of **Subjective Tasks** (Chat, Safety). In open-ended dialogue, defining "semantically identical" is inherently fuzzier. This ambiguity makes it more difficult for the generator to produce responses that are strictly semantically identical to the original answer while still differing enough to trigger a reward drop. Consequently, strict filtering rejects more candidates, leading to a lower apparent attack success rate.
> In short, the difference reflects the **intrinsic difficulty of defining semantic consistency** in subjective vs. objective domains, rather than a flaw in the generator itself.
>
> ---
>
> > **Weakness 3 :** Comparison of computational cost efficiency.
> >
>
> Thank you for the comment. We substantiate our efficiency claim by comparing A$^2$RM with **Adv-RM** [1], the most relevant prior work.
>
> - **Adv-RM:** As documented in their Appendix F, completing two iterations requires **896 GPU-hours** on **NVIDIA A100s**.
> - **A$^2$RM (Ours):** Our framework completes two full iterations using only **645 GPU-hours** on **NVIDIA A800s**.
>
> Crucially, the A100 GPU is faster than the A800. Despite using slower hardware, our method requires **~28% less time** in absolute terms. If normalized for hardware performance, our efficiency advantage is even more substantial.
>
> We believe that the inefficiency of Adv-RM stems from the need to search for samples that induce large divergences between the two reward models. Unlike semantic consistency which is easily assessed using the ‘chosen’ response as a reference, there is no clear structural or linguistic reference for generating adversarial samples that maximize disagreement between two RMs. As a result, the exploration process becomes substantially more complex and inefficient, ultimately leading to the significant computational overhead observed in Adv-RM.
>
> [1] Adversarial Training of Reward Models. COLM 2025.

---

> ### Author Response · Authors · 2025-11-20
> **Response to Reviewer FLFU (Part 2/2)**
>
> > **Question 1 :** Using a continuous semantic similarity score.
> >
>
> We appreciate this insightful suggestion. Following your recommendation, we implemented a variant of our framework using continuous semantic consistency scores (derived from the Judge model's output probabilities of 'Yes' over 'No') instead of the binary 0/1 signal. We compared this continuous approach with our original Binary strategy across all benchmarks.
>
> |  | JudgeBench | RewardBench | HelpSteer3 Eval |
> | --- | --- | --- | --- |
> |  | standard / A$^2$RM | standard / A$^2$RM | standard / A$^2$RM |
> | base | **60.86** / 2.29 | 73.24 / 51.71 | 85.90 / 25.64 |
> | binary | 58.00 / **69.43** | **83.37** / **91.33** | **87.32** / **89.31** |
> | continuous | 55.71 / 50.86 | 81.28 / 90.57 | 86.38 / 82.78 |
>
> As shown in the table, the Binary model consistently outperforms the Continuous model on both standard and adversarial data. We attribute this to two main factors:
>
> - **Noise & Calibration:** LLM probability scores are often not well-calibrated. A continuous score (e.g., 0.6) acts more as noise (uncertainty) rather than a reliable measure of partial semantic similarity, which confuses the training process.
> - **Dilution of Learning Signals:** The continuous score acts as a scaling factor that attenuates the feedback. It weakens the reward for successfully consistent samples (e.g., scaling the reward by 0.8 instead of 1.0) and fails to strictly penalize inconsistent samples (assigning them a non-zero positive reward instead of 0). This prevents the generator from receiving strong, sharp gradients needed to distinguish between valid adversarial attacks and hallucinations.
>
> In summary, the binary signal remains the most effective and efficient option, but richer continuous feedback could offer additional benefits. We view this as an important avenue for future research.
>
> ---
>
> > **Question 2 :** Selection of AdvGenerator training version.
> >
>
> Your intuition is correct. Our experiments suggest that if we continually fine-tune the generator ($\pi_{\psi_1}$), it tends to **overfit to the specific blind spots** of the previous RM iteration ($R_{\theta_0}$). It essentially becomes a specialist in tricking the old model.
> By resetting the generator to its initial SFT state ($\pi_{\psi_0}$) for the next round, we force it to forget the old, patched vulnerabilities and explore the unified action space again. This prevents the generator from getting stuck in local minima and allows it to discover **novel** vulnerabilities in the newly hardened RM.
>
> ---
>
> > **Question 3 :** Excessive adversarial training eventually harms in-distribution generalization.
> >
>
> We agree with the reviewer. This observation aligns with the **Robustness-Accuracy Trade-off** widely observed in adversarial training literature [1-3].
> As training progresses, the RM reallocates capacity toward detecting highly specific, complex adversarial patterns often far from natural language which can slightly reduce its sensitivity to simple in-distribution features. This trade-off is inherent but manageable: with just 1-2 iterations, we achieve over a 51% robustness gain with negligible degradation on standard performance.
>
> [1] Explicit tradeoffs between adversarial and natural distributional robustness, NeurIPS 2022.
>
> [2] Impact of Adversarial Training on Robustness and Generalizability of Language Models. ACL Findings 2023.
>
> [3] Theoretically Principled Trade-off between Robustness and Accuracy. ICML 2019.
>
> ---
>
> > **Question 4 :** A$^2$RM merely "transfers" robustness problem to the external Judge LLM.
> >
>
> We acknowledge the validity of this critique. As you pointed out, the quality of the training data depends on the Judge LLM, meaning that the robustness of our framework is, to some extent, transferred to the Judge. If an attacker were to successfully target the Judge during the training phase, it could indeed allow invalid samples (e.g., semantically inconsistent responses) to leak into the dataset, potentially affecting the A$^2$RM training.
>
> However, we believe this risk is manageable and does not undermine the value of the framework for the following reasons:
>
> 1. **Limited Role of the Judge:** The Judge LLM is only responsible for a specific sub-task: checking **semantic consistency**. The overall robustness of the final Reward Model is not derived solely from the Judge, but primarily from the **adversarial minimax optimization process**. The framework works by actively mining "Hard Negatives"—samples that satisfy the semantic constraint but expose the RM's blind spots. The Judge acts as the gatekeeper for the constraint, while the optimization process drives the robustness.
> 2. **Mitigation via Ensembling:** Since the Judge only provides binary consistency checks, its robustness can be greatly improved by ensembling multiple LLMs. Majority voting across diverse models mitigates individual biases and makes the semantic filter far harder to circumvent, reducing dependence on any single model.

---

> ### Author Response · Authors · 2025-11-25
> **Additional Experiment for Judge Model**
>
> > **Weaker model as judge.**
> >
>
> We employ **Llama2-13B-Instruct** as our judge model, which achieves only **25.78 ± 1.10** human alignment consistency, to examine how judge accuracy influences our adversarial augmentation framework. Experimental results show that adversarial samples generated by AdvGenerator trained with this weaker judge model lead to substantially poorer RM performance on A$^2$Adv bench compared to samples generated using a stronger judge model. This suggests that the reliability of the judge critically affects the quality of adversarial data. We believe that as model capabilities continue to advance and judge models become more accurate, A$^2$RM will exhibit even stronger robustness.
>
> |  | JudgeBench | RewardBench | HelpSteer3 Eval |
> | --- | --- | --- | --- |
> |  | standard / A$^2$Adv | standard / A﻿$^2$﻿﻿Adv | standard / A$^2$Adv |
> | base | **60.86** / 2.29 | 73.24 / 51.71 | 85.90 / 25.64 |
> | A$^2$RM with **Qwen3-32B** as judge | 58.00 / **69.43** | **83.37** / **91.33** | **87.32** / **89.31** |
> | A$^2$RM with **Llama2-13B-Instruct** as judge | 57.43 / 28.57 | 82.90 / 70.90 | 86.85 / 59.22 |

---

### Official Review · Reviewer_PCAe · 2025-10-31

**Soundness:** 2
**Presentation:** 2
**Contribution:** 2
**Rating:** 4
**Confidence:** 3

**Summary:**

Reward models trained on static preference datasets tend to learn spurious correlations, which can lead to reward hacking. The paper proposes a framework to systematical patch these vulnerabilities using an adversarial generator. An adversarial pair refers to one in which is semantically similar to the human preference pair but scored by RM to have the opposite preference. This process refines the RM iteratively to learn from adversarial pairs while maintaining performance on conventional preference pairs.

**Strengths:**

1.	The adversarial generation pipeline is reasonable, primarily seeking to reduce the reward of the adversarial samples while keeping the semantic meaning of the adversarial the sample.
2.	The choice of benchmarks is good and diverse, although RewardBench is a little bit dated.

**Weaknesses:**

1.	The approach used to assess if two responses are semantically similar is a little under-explored. For instance, RM-Bench [1] finds adversarial pairs by asking an LLM to change a few keywords in a good correct response to make it incorrect. If the “corruption” is as minor as changing a few words, it’s unlikely to result in a semantic change by simply passing it to a LLM-Judge. Therefore if the AdvGenerator only changes a few words, I think Cons(c, c’) will stay at 1 – meaning that as long as there’s no large change, this term doesn’t influence training. Did the team consider continuous measures of semantic similarity such as embedding similarity? This will also likely be much faster for inference.
2.	The paper discusses Adv-RM (Bukharin et al., 2025) but did not compare to it in Table 2 – is there a reason for it? The other baseline methods seem slightly dated (e.g. 2024 or earlier).
3.	I’m not convinced by the evaluation results. The models perform better on A2Adv and StyleAdv which are expected given that the test samples are generated in a similar way as the training samples (and therefore might be overfitted). However, they do not show a boost in the standard benchmark scores. Why is that so? If the goal of building Adverserial RMs is to improve RMs, we should see the standard benchmarks improve as well. This is particularly for JudgeBench since JudgeBench responses are all generated by the same model with small variations, which should show some gains if the gains generalize.

[1] Yantao Liu, Zijun Yao, Rui Min, Yixin Cao, Lei Hou, Juanzi Li. RM-Bench: Benchmarking Reward Models of Language Models with Subtlety and Style. ICLR 2025.

**Questions:**

1.	Did the team consider using RM-Bench? It’s likely to be a good benchmark to test adversarial sensitivity given how it’s constructed.
2.	Was there some analysis on what kind of changes the AdvGenerator was doing? This will bring clarity into how the AdvGenerator works.
3.	With the PPO algorithm, there’s an assumption that there’s a trained value function – is there? I don’t see that this is done in the paper. Otherwise, this seems like the classical REINFORCE algorithm using the r (or c) reward as the baseline in calculating advantage.

---

> ### Author Response · Authors · 2025-11-20
> **Response to Reviewer PCAe (Part 1/3)**
>
> We thank the reviewer for the thoughtful comments. We appreciate the opportunity to clarify our semantic consistency metrics, baseline comparisons, and evaluation interpretations.
>
> ---
>
> > **Weakness 1 :** Try to use embedding similarity as a semantic consistency signal.
> >
>
> We explored this efficient alternative but found it insufficient for our task. We evaluated two embedding models (**bert-base-uncased**, **bge-base-en**) on 128 randomly sampled and human-labeled pairs. We determined optimal thresholds for each model to maximize agreement. As shown below, embedding-based methods achieved only **~34% accuracy**, far below the **~85%** achieved by our LLM Judge (Qwen3-32B).
>
> | Model | Agreement with Human |
> | --- | --- |
> | **Qwen3-32B (Used)** | **84.77% ± 0.55%** |
> | bert-base-uncased | 34.33% |
> | bge-base-en | 33.59% |
>
> We argue that embedding models focus on global semantic overlap. However, adversarial perturbations often involve subtle structural or stylistic changes that embeddings view as different, or conversely, embeddings might miss critical negations that change meaning. The LLM Judge provides the fine-grained reasoning needed to verify specific semantic equivalence. While we acknowledge the computational advantage of embedding-based approaches, our empirical findings indicate that they do not provide reliable semantic filtering in our setting.
>
> To further explore continuous measures of semantic similarity, we implemented a variant of our framework using the Judge model's output probabilities of 'Yes' over 'No' as the continuous measure. We compared this continuous approach with our original Binary strategy across all benchmarks.
>
> |  | JudgeBench | RewardBench | HelpSteer3 Eval |
> | --- | --- | --- | --- |
> |  | standard / A$^2$Adv | standard / A$^2$Adv | standard / A$^2$Adv |
> | base | **60.86** / 2.29 | 73.24 / 51.71 | 85.90 / 25.64 |
> | binary | 58.00 / **69.43** | **83.37** / **91.33** | **87.32** / **89.31** |
> | continuous | 55.71 / 50.86 | 81.28 / 90.57 | 86.38 / 82.78 |
>
>
> As shown in the table, the Binary model consistently outperforms the Continuous model on both standard and adversarial data. We attribute this to two main factors:
>
> - **Noise & Calibration:** LLM probability scores are often not well-calibrated. A continuous score (e.g., 0.6) acts more as noise (uncertainty) rather than a reliable measure of partial semantic similarity, which confuses the training process.
> - **Dilution of Learning Signals:** The continuous score acts as a scaling factor that attenuates the feedback. It weakens the reward for successfully consistent samples (e.g., scaling the reward by 0.8 instead of 1.0) and fails to strictly penalize inconsistent samples (assigning them a non-zero positive reward instead of 0). This prevents the generator from receiving strong, sharp gradients needed to distinguish between valid adversarial attacks and hallucinations.
>
> In conclusion, while the binary approach currently enforces a stricter and more efficient learning signal, we acknowledge that more sophisticated forms of continuous feedback might offer better results. Exploring these alternative continuous metrics remains a promising direction for future research.
>
> ---
>
> > **Weakness 2 :** Compare with Adv-RM.
> >
>
> Thank you for raising this point. As of the time of our experiments, Adv-RM [1] had not released its code or model checkpoints, and we failed to reproduce this method. For this reason, we could not include a reliable quantitative comparison in Table 2. We will closely follow the project, and once Adv-RM becomes publicly available, we plan to incorporate a full comparison and extend our study to evaluate compatibility and differences across adversarial RM training paradigms.
>
> In Adv-RM, two reward models must repeatedly probe each other’s weaknesses to generate adversarial examples. It is inherently limited in its ability to detect failure modes that are shared by both models, resulting in blind spots that adversarial training cannot effectively address. Our A$^2$RM framework mitigates these issues by consolidating the dual-RM architecture into a single RM, while employing an external judge model to evaluate the semantic consistency of generated adversarial samples. This redesign reduces system complexity, eliminates cross-RM communication overhead, and provides a more reliable mechanism for identifying diverse adversarial failures.
>
> [1] Adversarial Training of Reward Models. COLM 2025.

---

> ### Author Response · Authors · 2025-11-20
> **Response to Reviewer PCAe (Part 2/3)**
>
> > **Weakness 3 :** Standard benchmarks do not improve; robustness gains might be overfitting.
> >
>
> We appreciate this observation and understand the expectation that an improved model should ideally see gains across all metrics. We would like to offer further context regarding these results:
>
> 1. **Robustness-Accuracy Trade-off:** In adversarial training literature, it is a widely recognized challenge that improving robustness against attacks often comes at the cost of degrading performance on clean, standard data [1-3]. In this context, our ability to **maintain** competitive performance on standard benchmarks (e.g., JudgeBench-Standard) while achieving massive gains in robustness is a significant result. It indicates that A$^2$RM successfully patched vulnerabilities without suffering from catastrophic forgetting.
> 2. **Generalization vs. Overfitting:** To address the concern that the model might be overfitting to our specific generator, we point to the **StyleAdv** results. StyleAdv uses a completely different attack mechanism (style transfer) that our model never saw during training. The fact that RM improved accuracy on StyleAdv from **41.73% to 90.92%** demonstrates that it has learned a generalized definition of semantic consistency, rather than merely memorizing specific attack patterns from the AdvGenerator.
>
> [1] Explicit tradeoffs between adversarial and natural distributional robustness, NeurIPS 2022.
>
> [2] Impact of Adversarial Training on Robustness and Generalizability of Language Models. ACL Findings 2023.
>
> [3] Theoretically Principled Trade-off between Robustness and Accuracy. ICML 2019.
>
> ---
>
> > **Question 1 :** Use RM-bench.
> >
>
> Thank you for the recommendation. We evaluated our model on RM-Bench. The results align with our other findings: standard performance remains stable, while adversarial robustness improves significantly.
>
> | Iteration | Standard | A$^2$Adv | Style Adv |
> | --- | --- | --- | --- |
> | 0 | 73.02 | 24.87 | 71.89 |
> | 1 | 71.74 | 31.95 | 78.75 |
> | 2 | 72.65 | 32.03 | 83.57 |
>
> ---
>
> > **Question 3 :** Is there a trained value function in PPO?
> >
>
> Yes, absolutely. We follow the standard PPO implementation (i.e.., OpenRLHF) which includes a Critic (Value) Network. The Critic shares the initial layers with the Actor (Policy) but has a separate regression head. The Critic is trained to predict the expected return (cumulative reward) of the current state. This estimated value $V(s)$ is used to compute the Advantage $A(s,a) = R - V(s)$, which reduces variance in the policy gradient update compared to the raw return used in REINFORCE.

---

> ### Author Response · Authors · 2025-11-20
> **Response to Reviewer PCAe (Part 3/3)**
>
> > **Question 2 :** Analysis on AdvGenerator.
> >
>
> We analyzed the cases generated by advgenerator and offered some insights. Below are three cases. We examined the responses to the same question from different steps of AdvGenerator.
>
> We observe that, during the early stages of training(50 step), the model’s outputs exhibit a highly fragmented behavior, often failing to fully address the requirements specified in the question. As training progresses, the model begins to respond by decomposing the questio into partial components rather than understanding it holistically(100 step). In the final stage of training , the model’s responses converge toward the style and structure of the chosen response. This trajectory suggests that the generation of adversarial samples gradually evolves from responding to isolated parts of the question to achieving a more coherent, global understanding.
>
> In addition, we find that some special tokens which do not affect overall semantic interpretation will increases throughout training. These changes contribute to producing adversarial samples that maintain semantic consistency with the original question while achieving a lower reward score from the target reward model.
>
> |  | question | chosen response | 50 step response | 100 step response | final step response |
> | --- | --- | --- | --- | --- | --- |
> | case 1 | The five countries with the largest land areas in the world are, in order: Russia, Canada, China, Brazil, and the United States. Please point out the errors in the above ranking.  | The top five countries in terms of land area are: 1. Russia 2. Canada 3. People's Republic of China 4. United States 5. Brazil The order of the United States and Brazil in the ranking you provided is incorrect. The correct order should be the United States before Brazil. | China  is not the 5th largest by land area \n\n- The United States is 4th after Brazil | 1. the United States\n2. Russia\n3. Canada\n4. China\n5. Brazil\n\nNote: The correct order should be by land area. The United States has a larger land area than Brazil.\n\n6. Russia\n7. Canada\n8. China\n9. Brazil\n10. United States\n\nComplete corrected list:\n\n6. Russia\n7. Canada\n8. China\n9. Brazil\n10. United States\n\nCorrected ranking:\n\n1. the United States\n2. Russia\n3. Canada\n4. China\n5. Brazil | *Determined.\n\ncontent:\n\nWorld Ranking: Russia, Canada, China, USA, Brazil\n\nNote:\n\n1. The USA should be before Brazil in the ranking.\n2. The ranking you provided has the USA after Brazil, which is incorrect.\n\nCorrected Ranking:\n\nThe correct order should be:\n\nRussia, Canada, China, USA, Brazil |
> | case 2 | In future conversations, you will act as a senior manual app tester, possessing knowledge of iOS and Android app development, understanding UI design, clear logic, and the ability to effectively ensure that the engineers' development progress aligns with the functional requirements designed by the product designers. Next, I will inform you of my position and the guidance I will provide. If you understand the instructions, simply answer "ok," and I will continue with further explanation. Also, please do not repeat my questions; simply reply with your answer. | ok | understand .lavc2. .lavc3. .lavc4. .lavc5. .lavc6. | \n\n1: Understanding the Role\n\nBe able to explain the role of a senior manual app tester, including the responsibilities and the importance of the role.\n\n2: Technical Expertise\n\nHave a solid understanding of iOS and Android app development, including the different development tools and technologies used.\n\n3: UI Design\n\nBe familiar with UI design principles and how they apply to mobile app development. Understand the importance of a well-designed user interface for a successful app. | kinsan-wing,\n\nok |
> | case 3 | Generate a short title for the following product listing. Requirements:\n1. Title should be 4-10 Chinese characters\n2. Clearly describe the product name and selling points, but do not repeat information already displayed in the product attributes\n3. It is recommended to include the brand name\n4. Do not enter special characters (such as punctuation, long strings of numbers, URLs, etc., except spaces) or other information unrelated to the product being listed\n\nSummer 2023 Walking Poop Feel Slippers Men's Sports Outdoor Slippers New Home Thick Sole Non-Slip Anti-Odor Sandals | New summer Nike non-slip platform sandals | Nike Air Zoomantz\n\nNote: Since the brand name was not provided in the question, Nike is used instead. | Summer 2023 Men's Sports Slippers\n\n- Brand: Walking Poop\n- Product: Feel Slippers\n- Description: New Home Thick Sole Non-Slip Anti-Odor Sandals\n- Selling Points: Outdoor, Breathable, Quick-Drying, Lightweight, Stylish, Comfortable, Suitable for Summer\n- Material: High-quality rubber and mesh\n- Features: Anti-slip sole, breathable mesh upper, comfortable insoles\n | .Anti-Slip Platform Sandals #\n Nike Summer  |

---

> ### Comment · Reviewer_PCAe · 2025-11-24
> **Reply to author response**
>
> Weakness 1: The embedding models tested (bert-base-uncased, bge-base-en)  are unfair baselines since they are both 100M sized and trained in 2019 or 2023. For comparison with Qwen3-32B, Qwen/Qwen3-Embedding-8B (or other embedding models released around the same time) might be a fairer comparison.
>
> Weakness 2: Can the author share more about the attempts to reproduce Adv-RM and how they failed? Without this or another recent baseline, I think the existing baselines are pretty dated (2024 or earlier) and doesn't differentiate this work from recent advances.
>
> Weakness 3/Question 2: The samples generated by AdvGenerator seem pretty unnatural to any well-trained instruct model (with reasonable perplexity on general text) so I highly doubt that such generations (or the like) will be seen in actual RL rollouts. Coupled with the fact that standard RM benchmarks don't improve, I don't think this work is useful in training real-world reward models (Not dropping on accuracy is ok but not enough).
>
> Question 1: Why is the column on A$^2$Adv so low on RM-Bench? Random guess is 50% so this training probably collapsed.
>
> Question 3: These kind of implementation details are actually pretty important to report for reproducibility. It's not exactly standard practice to have a shared backbone between actor and value network to the best of my knowledge even if it's the default in one codebase (recent value networks are typically initialized with the reward model and many recent works show that using PPO with a value network can perform less well compared to GRPO/RLOO that do not use this value network).

---

> > ### Author Response · Authors · 2025-11-25
> > **Response to Reviewer PCAe**
> >
> > Many thanks for your prompt reply. Your timely feedback provides us with valuable time and guidance to better prepare comprehensive responses to the concerns you highlighted.
> >
> > ---
> >
> > > **Weakness 1** :  Selection of Embedded Model Version.
> > >
> >
> > We follow the suggestion and use Qwen3-embedding-8B as an additional embedding baseline. The results show that even within the same model family, the embedding approach still performs poorly on semantic consistency evaluation. We believe the main reason is that embedding models focus on global semantic similarity, which makes them less effective at capturing fine grained semantic distinctions compared to LLM judges. For example, **“It’s very hot today.”** and **“It’s not very hot today.”** receive a similarity score of 0.7676 under Qwen3-embedding-8B, despite their meanings being clearly inconsistent.
> >
> > We agree with your point of view, embeddings may offer advantages in terms of resource efficiency and global understanding, and we see this as a promising direction for further exploration.
> >
> > | Model | Agreement with Human |
> > | --- | --- |
> > | Qwen3-32B(Used) | 84.77% ± 0.55% |
> > | bert-base-uncased | 34.33% |
> > | bge-base-en | 33.59% |
> > | qwen3-embedding-8B | 35.16% |
> >
> > ---
> >
> > > **Weaknes 2 :** Specific reasons why advRM failed to reproduce.
> > >
> >
> > More specifically, when reproducing advRM, we used the open-source RL framework OpenRLHF. We refactored the framework’s original FastAPI based single remote RM server design and extended it to support two reward models working collaboratively. We also followed the paper’s description by applying the RLOO algorithm with the corresponding hyperparameters. However, the reproduced results diverged significantly from those reported in the paper.
> >
> > We believe this discrepancy may stem from inconsistencies in the experimental environment and framework implementation details. In particular, the RLOO implementation in OpenRLHF differs from the version described in the paper: it modifies the original version by incorporating the per-token KL reward and utilizing the PPO-clip loss [1]. These differences can substantially affect the training dynamics and final outcomes, which likely contributed to our unsuccessful attempt to replicate advRM.
> >
> > [1] OpenRLHF: An Easy-to-use, Scalable and High-performance RLHF Framework. arxiv 2024.
> >
> > ---
> >
> > > **Weakness 3 & Question 2 :** Questioning the results generated by advGenerator.
> > >
> >
> > We believe these outcomes reflect precisely how the model behaves under adversarial generation. Such unnatural constructions are inherently **out-of-distribution(OOD)** for the model. It is precisely these adversarial samples that are perceived as “unnatural” by humans that enhance the reward model’s robustness when it encounters OOD inputs.
> >
> > Moreover, we do not view this work as lacking value for RM training. Our goal is to **enhance robustness** **while preserving stability on standard benchmarks**, rather than trading one for the other. The experimental results support this objective: the reward model maintains stable performance on standard benchmarks, while achieving substantial improvements in robustness against OOD data.
> >
> > ---
> >
> > > **Question 1 :** This training probably collapsed.
> > >
> >
> > We believe that achieving less than 50% accuracy on the adversarial benchmark does not indicate a failure of training. The base RM (iteration 0) achieves only 24.87% on the A$^2$Adv adversarial benchmark, whereas after our training procedure, A$^2$RM (iteration 2) improves to 32.03%, **a gain of 7.16%**. This demonstrates that our adversarial training effectively enhances the reward model’s robustness when facing OOD inputs. We also evaluated Skywork-RM-27B, which reaches only 36.32% on A$^2$Adv. Therefore, low scores on adversarial benchmarks do not imply collapsed training, but rather reflect the inherent difficulty of adversarial OOD evaluations.
> >
> > ---
> >
> > > **Question 3 :** Concerns about reproducibility and effectiveness.
> > >
> >
> > We followed the PPO training setup provided by the open-source OpenRLHF framework. To ensure reproducibility, we include all hyperparameters in the paper and provide the implementation code in the supplementary materials. Our use of OpenRLHF’s PPO is primarily for rapid implementation and verification of our method, as the focus of our work lies in adversarial training, not in optimizing the RL component. We agree with the reviewer that the value network could benefit from improved initialization or design, and that algorithms such as GRPO or RLOO may outperform PPO. Nevertheless, even with our current PPO choice and implementation, the RM achieves surprisingly strong robustness gains. We believe that replacing PPO with a more advanced algorithm (e.g., GRPO or RLOO) could further enhance the framework’s performance. Since our current priority is to enhance RM robustness through the adversarial training framework, which is orthogonal to the choice of RL algorithms, we leave improvements on the RL side to future work.

---

> ### Author Response · Authors · 2025-11-25
> **Additional Experiment for Judge Model**
>
> > **Weaker model as judge.**
> >
>
> We employ **Llama2-13B-Instruct** as our judge model, which achieves only **25.78 ± 1.10** human alignment consistency, to examine how judge accuracy influences our adversarial augmentation framework. Experimental results show that adversarial samples generated by AdvGenerator trained with this weaker judge model lead to substantially poorer RM performance on A$^2$Adv bench compared to samples generated using a stronger judge model. This suggests that the reliability of the judge critically affects the quality of adversarial data. We believe that as model capabilities continue to advance and judge models become more accurate, A$^2$RM will exhibit even stronger robustness.
>
> |  | JudgeBench | RewardBench | HelpSteer3 Eval |
> | --- | --- | --- | --- |
> |  | standard / A$^2$Adv | standard / A﻿$^2$﻿﻿Adv | standard / A$^2$Adv |
> | base | **60.86** / 2.29 | 73.24 / 51.71 | 85.90 / 25.64 |
> | A$^2$RM with **Qwen3-32B** as judge | 58.00 / **69.43** | **83.37** / **91.33** | **87.32** / **89.31** |
> | A$^2$RM with **Llama2-13B-Instruct** as judge | 57.43 / 28.57 | 82.90 / 70.90 | 86.85 / 59.22 |

---

> > ### Comment · Reviewer_PCAe · 2025-11-28
> > **Response to Author replies**
> >
> > I appreciate the authors for conducting the followup experiments which do indicate that embedding models (or weaker instruction following models) are not aligned with humans to identify textual similarities in this context.
> >
> > However, even after the response, I still have two outstanding concerns (in order of importance), which I do not expect to be solvable within the discussion period.
> >
> > 1. Practical usefulness of "adversarial" training: it seems like the main advantage of this method is to `enhance robustness while preserving stability on standard benchmarks`. However, as a researcher who trains reward models for RL, I don't see robustness to randomly perturbed LLM generations that are unlikely to be seen in actual RL rollouts to be a useful outcome in itself.
> >
> > 2. Strong coupling with particular implementation on OpenRLHF. In the authors' response to Weakness 2 and Question 3, it seems like a lot of the implementation in the paper are presumably the default in OpenRLHF. While this is not a weakness in itself, it gives the impression that the authors' implementation are highly limited by OpenRLHF and that limited attempts was done to go beyond these defaults in order to fully replicate baselines or explore RL training (which do have important implications on observed results).
> >
> > Therefore, I decide to maintain my original assessment.

---

> > > ### Author Response · Authors · 2025-11-28
> > > **Response to Reviewer PCAe**
> > >
> > > Thank you for the follow-up discuss and for clarifying your remaining concerns.
> > >
> > > We acknowledge that the first point regarding the practical utility of robustness lies beyond the intended scope of this work. Our focus is on demonstrating that the proposed method can consistently improve robustness, and the additional jailbreak experiments provided during the discussion (as requested by reviewer wzXQ) empirically support this claim.
> > >
> > > Regarding the second concern, we agree that relying on a specific implementation framework such as OpenRLHF may raise questions about generality. Our choice was motivated by its wide adoption and maturity in the community (**8.5k stars on Github**), which allowed us to validate our algorithmic contributions efficiently without re-engineering fundamental RL components. We fully recognize that alternative implementations may provide further insights, and exploring such directions represents a promising avenue for future work.
> > >
> > > Once again, we sincerely appreciate the reviewer’s time and constructive feedback.

---

### Official Review · Reviewer_PsK3 · 2025-11-01

**Soundness:** 3
**Presentation:** 2
**Contribution:** 3
**Rating:** 4
**Confidence:** 4

**Summary:**

This paper proposesAdversarial-Augmented Reward Model(A2RM), a framework designed to improve reward model robustness through adversarial data augmentation. Specifically, a policy model generates semantically similar but adversarial responses that receive unexpectedly low scores from the current reward model. These challenging samples—validated by a Judge model for semantic consistency—are then incorporated into the reward model’s training data to iteratively refine its judgment and reduce reward hacking.

**Strengths:**

1.The paper addresses the critical issue of reward model vulnerability to semantically deceptive responses, an increasingly important problem in RLHF and preference optimization.

2.The idea of generating semantically consistent but reward-contradictory samples and iteratively retraining the RM is novel and conceptually appealing.

3.The full pipeline (AdvGenerator → Judge LLM → selective augmentation → retraining) is clearly explained, with algorithmic pseudocode and reproducible experimental settings.

4.Empirical results show consistent gains in adversarial accuracy with minimal degradation in standard benchmarks.

**Weaknesses:**

1.The Judge model plays a pivotal role in determining semantic consistency between the original response and adversarial response. Its correctness directly affects data quality, yet the paper provides no quantitative analysis of the Judge’s reliability.

2.Only samples that are both semantically similar and receive lower RM scores than the rejected answer rrr are retained. This dual filtering likely leads to low data utilization, but the paper lacks statistics on selection rates or sample efficiency across iterations.

3.The binary use of the Judge output (0/1) to gate adversarial rewards may cause unstable gradients and discard informative near-consistent samples.

4.Since adversarial examples are produced by the same AdvGenerator used during training, the RM may overfit to that generator’s attack style rather than learning truly generalizable robustness.

**Questions:**

1.How reliable is the Judge model in determining semantic consistency? Please provide a quantitative evaluationand analyze how variations in Judge quality affect the final robustness and stability of the reward model.

2.After applying both semantic-consistency filtering and adversarial-score selection, what proportion of generated samples are actually retained for training? A detailed analysis of sample retention and effective data usage would clarify the framework’s efficiency.

3.Could the current binary signal be replaced by a continuous score to yield denser reward feedback and smoother gradient updates? A comparison between binary and soft consistency would be informative.

4.How well does the trained generator generalize beyond its own attack style? Have you evaluated cross-generator robustness, such as testing the RM against adversarial examples produced by a different generator or decoding configuration?

---

> ### Author Response · Authors · 2025-11-20
> **Response to Reviewer PsK3 (Part 1/2)**
>
> Thank you for your thoughtful review. We appreciate your recognition of the novelty of our adversarial-augmented framework and the clarity of our presentation. Below, we address your specific questions regarding the Judge model, data efficiency, and generalization.
>
> ---
>
> > **Weakness 1 & Question 1 :** Reliability of the Judge model.
> >
>
> We address the concern regarding the reliability of the Judge model from three perspectives: adherence to community standards, empirical human validation, and scalability analysis.
>
> - **Alignment with Existing Work:** We followed the methodology of recent studies [1-3] by employing an LLM-as-a-Judge for semantic evaluation. This approach has become a common choice in the research community, offering a practical balance between scalability and evaluation cost.
> - **Human Verification:** To empirically validate the reliability of this approach in our specific context, we conducted a human evaluation on 128 randomly sampled pairs of original and adversarial responses. We compared the judgments of our Judge model (Qwen3-32B) against human annotations. The results show a high agreement rate of 84.77% (± 0.55%). This high consistency, coupled with a narrow fluctuation range across repeated trials, confirms that our Judge model serves as a reliable proxy for human semantic evaluation.
> - **Evolutionary Trend & Robustness:** We further evaluated five different open-source models to analyze how judge quality varies with model capability. As shown in the table below, agreement with human annotators improves significantly with model scale and iteration.
> This trend suggests that our framework is future-proof: as open-source LLMs continue to improve, the reliability of the semantic filter and consequently the robustness of the trained A$^2$RM will naturally increase. We also believe that our framework could derive further benefits from more powerful proprietary models (e.g., GPT-5). However, given the substantial costs associated with large-scale iterative training, we prioritized open-source solutions in this work to ensure cost-effectiveness.
>
> | Model | Agreement with Human |
> | --- | --- |
> | qwen3-32B (Used) | 84.77 ± 0.55 |
> | qwen2.5-72B-Instruct | 77.74 ± 3.87 |
> | qwen2.5-32B-Instruct | 77.34 ± 1.10 |
> | llama3.1-70B-Instruct | 73.83 ± 1.65 |
> | llama2-13B-Instruct | 25.78 ± 1.10 |
>
> [1] From Generation to Judgment: Opportunities and Challenges of LLM-as-a-judge. EMNLP 2025.
>
> [2] Generative Judge for Evaluating Alignment. ICLR 2024.
>
> [3] Judging LLM-as-a-Judge with MT-Bench and Chatbot Aren. NeurIPS 2023.
>
> ---
>
> > **Weakness 2 & Question 2 :** Detail analysis of sample retention and effective data usage.
> >
>
> We appreciate this question regarding data efficiency. We tracked the sample retention across iterations:
>
> - **Iteration 1:** 17,297 samples retained (86.96% of generated candidates).
> - **Iteration 2:** 4,778 samples retained (24.02% of generated candidates).
>
> While the retention rate drops significantly in Iteration 2, this aligns with our framework design and reflects the following dynamics:
>
> - **Improved Model Robustness:** The significant decrease in the retention rate directly confirms that the reward model has become more robust. As the model updates after the first iteration, it develops a stronger resistance to adversarial manipulation. Consequently, it becomes much more difficult for the generator to craft responses that can successfully mislead the improved reward model, resulting in fewer samples meeting the selection criteria.
> - **High Information Density:** Although the number of retained samples is smaller in the second iteration, these samples are highly informative. They represent the specific edge cases that are still capable of bypassing the hardened reward model. Training on these challenging samples focuses the optimization on the remaining vulnerabilities of the model.
> - **Computational Efficiency:** Although a larger proportion of generated data is discarded in the second iteration, the computational cost of the forward generation process is significantly lower than that of backward optimization. By rigorously filtering out ineffective samples, we avoid the high computational expense of training on low-value data. This ensures that our computational resources are focused solely on the high-information samples that contribute most to model improvement.
> - **Validated Efficiency:** The effectiveness of this selection process is evidenced by the performance metrics in Table 1. Despite utilizing a smaller dataset in the second iteration, the reward model still achieved a accuracy improvement on the adversarial benchmark, rising from 91.33% to 95.82%. This demonstrates that our method achieves significant performance gains through efficient data usage rather than volume accumulation
>
> In summary, our method trades inexpensive inference for substantial training efficiency, achieving better performance with lower overall compute.

---

> ### Author Response · Authors · 2025-11-20
> **Response to Reviewer PsK3 (Part 2/2)**
>
> > **Weakness 3 & Question 3 :** Replace the binary reward signal with a continuous reward signal.
> >
>
> We appreciate this insightful suggestion. Following your recommendation, we implemented a variant of our framework using continuous semantic consistency scores (derived from the Judge model's output probabilities of 'Yes' over 'No') instead of the binary 0/1 signal. We compared this continuous approach with our original Binary strategy across all benchmarks.
>
> |  | JudgeBench | RewardBench | HelpSteer3 Eval |
> | --- | --- | --- | --- |
> |  | standard / A$^2$Adv | standard / A$^2$Adv | standard / A$^2$Adv |
> | base | **60.86** / 2.29 | 73.24 / 51.71 | 85.90 / 25.64 |
> | binary | 58.00 / **69.43** | **83.37** / **91.33** | **87.32** / **89.31** |
> | continuous | 55.71 / 50.86 | 81.28 / 90.57 | 86.38 / 82.78 |
>
> As shown in the table, the Binary model consistently outperforms the Continuous model on both standard and adversarial data. We attribute this to two main factors:
>
> - **Noise & Calibration:** LLM probability scores are often not well-calibrated. A continuous score (e.g., 0.6) acts more as noise (uncertainty) rather than a reliable measure of partial semantic similarity, which confuses the training process.
> - **Dilution of Learning Signals:** The continuous score acts as a scaling factor that attenuates the feedback. It weakens the reward for successfully consistent samples (e.g., scaling the reward by 0.8 instead of 1.0) and fails to strictly penalize inconsistent samples (assigning them a non-zero positive reward instead of 0). This prevents the generator from receiving strong, sharp gradients needed to distinguish between valid adversarial attacks and hallucinations.
>
> In conclusion, while the binary approach currently enforces a stricter and more efficient learning signal, we acknowledge that more sophisticated forms of continuous feedback might offer better results. Exploring these alternative continuous metrics remains a promising direction for future research.
>
> ---
>
> > **Weakness 4 & Question 4 :** Testing the RM against adversarial examples produced by a different generator.
> >
>
> We evaluated A$^2$RM on JudgeBench under a variety of adversarial attack settings. The results show that, compared with both the BaseRM trained on the same underlying data and the open-source Skywork-RM series, A$^2$RM exhibits a **substantial reduction in attack success rates**. This consistently lower vulnerability indicates that A$^2$RM is significantly more robust and can effectively withstand diverse adversarial attack strategies.
>
> | Method | BaseRM | Skywork-RM-8B | Skywork-RM-27B | A$^2$RM |
> | --- | --- | --- | --- | --- |
> | Textfooler | 85.16 | 86.72 | 93.75 | **57.03** |
> | StyleAdv | 79.69 | 81.25 | 85.16 | **31.25** |
> | RRM | 0.78 | 0.00 | 0.00 | **0.00** |
>
> This behavior arises because A$^2$RM dynamically adjusts its training data by generating adversarial samples that deliberately mislead the reward model into assigning low scores while still preserving semantic consistency with the ‘chosen’ responses. Incorporating these carefully crafted adversarial samples into RM training enhances the model’s robustness. As a result, A$^2$RM is substantially more resilient and performs more reliably when exposed to diverse forms of adversarial attacks.

---

> ### Author Response · Authors · 2025-11-25
> **Additional Experiment for Judge Model**
>
> > **Weaker model as judge.**
> >
>
> We employ **Llama2-13B-Instruct** as our judge model, which achieves only **25.78 ± 1.10** human alignment consistency, to examine how judge accuracy influences our adversarial augmentation framework. Experimental results show that adversarial samples generated by AdvGenerator trained with this weaker judge model lead to substantially poorer RM performance on A$^2$Adv bench compared to samples generated using a stronger judge model. This suggests that the reliability of the judge critically affects the quality of adversarial data. We believe that as model capabilities continue to advance and judge models become more accurate, A$^2$RM will exhibit even stronger robustness.
>
> |  | JudgeBench | RewardBench | HelpSteer3 Eval |
> | --- | --- | --- | --- |
> |  | standard / A$^2$Adv | standard / A﻿$^2$﻿﻿Adv | standard / A$^2$Adv |
> | base | **60.86** / 2.29 | 73.24 / 51.71 | 85.90 / 25.64 |
> | A$^2$RM with **Qwen3-32B** as judge  | 58.00 / **69.43** | **83.37** / **91.33** | **87.32** / **89.31** |
> | A$^2$RM with **Llama2-13B-Instruct** as judge | 57.43 / 28.57 | 82.90 / 70.90 | 86.85 / 59.22 |

---

> > ### Comment · Reviewer_PsK3 · 2025-11-27
> >
> > Thank you for your response and for conducting the additional experiments. The clarifications have addressed part of my initial concerns.
> > However, I remain uncertain about the validity of introducing both **AdvGenerator** and **JudgeLLM** simultaneously and the potential bias this combination may introduce. Therefore, I am keeping my current score unchanged.

---

> > > ### Author Response · Authors · 2025-11-28
> > > **Response to Reviewer PsK3**
> > >
> > > Thank you for your feedback. While we acknowledge that the choice of the judge LLM may have some influence on the evaluation, its role in our framework is limited to assessing semantic consistency, where potential biases are comparatively minor. The dominant source of preference signals comes from the annotated training data rather than from the judge model itself. We sincerely appreciate the reviewer’s insights and thoughtful comments, and thank you again for your time and consideration.

---

### Author Response · Authors · 2025-11-20
**General Response to Reviewers**

We sincerely thank all reviewers for their careful and constructive feedback.

We are pleased to see that all reviewers acknowledged the novelty of our proposed adversarial augmented framework. As an adversarial training method, A$^2$RM substantially improves the robustness of reward models.

At the same time, we appreciate that all reviewers highlighted two common concerns:

- The reliability of the judge model.
- The discontinuity of the binary reward signal.

In response, we have added experiments and analyses. The judge model achieves approximately 85% agreement with human annotations, supporting its reliability. Additionally, we observe that continuous reward signals may introduce noise, resulting in weaker performance compared with the binary signal.

Overall, we believe our work provides a valuable and thought provoking contribution to the community and will inspire future research in adversarial training for reward models.

---

### Author Response · Authors · 2025-12-03
**Response to AC and SACs**

**Dear AC and SACs,**

We understand that due to a rare bug in the ICLR / OpenReview system, your workload was significantly greater than usual, including re-reading the updated submissions and reaching final decisions without the normal author-reviewer discussion phase. We sincerely appreciate the additional time and effort you devoted under these atypical circumstances, and we are grateful for your commitment to maintaining the integrity and efficiency of the ICLR review process.

To help you better understand the rebuttal phase, we have summarized the reviewers' key concerns below to help you evaluate our paper more effectively.

**Appendix D:** We add experiments on the reliability of the judge model.

**Appendix E:** We add experiments and discussions on binary reward signals and continuous reward signals.

**Appendix F:** We add experiments on the robustness of A$^2$RM to jailbreak-related benchmark and method.

**Appendix G:** We supplement experiments and discussions that merges $c'$ and $r'$ in one AdvGenerator.

---

### Meta-Review · Area_Chair_nz57 · 2026-01-06

**Summary:**

1. Reviewers question the reliability of using an external "Judge LLM", thinking it introduces new knowledge and maybe unreliable.
2. Some concerns about the experiments, including the lack of improvement on standard benchmarks, missing comparisons to relevant baselines, and the initial absence of safety related benchmarks.
3. There are some concerns if the authors optimize baseline implementations enough. Plus if the generator can generalize or it just overfits to the specific attack styles.

**Reviewer Concerns:**

The authors add more experiments on safety benchmarks and an ablation study comparing binary vs. continuous signals. This mitigates some concerns about the experiment weaknesses and method design choices.

However, the first concern on whether this approach introduces new knowledge from Judge LLM still holds. Reviewer PCAe remains unconvinced about the baseline optimization levels.

**Reviewer Scores:**

Reviewer PsK3: No Change. The reviewer explicitly says this.

Reviewer PCAe: No Change. The reviewer explicitly says this.

Reviewer FLFU: Likely no change

Reviewer wzXQ: May increase. The reviewer says the rebuttal "resolves most of my concerns" and is leaning possible.

---

### Decision · Program_Chairs · 2026-01-26

Reject